# Empowering the on-site detection of nucleic acids by integrating CRISPR and digital signal processing

Chang Yeol Lee [1,2,3,10], Hyunho Kim [1,2,10], Ismail Degani[1,4,10], Hanna Lee[1], Angel Sandoval [1], Yoonho Nam [1,5], Madeleine Pascavis [1,6], Hyun Gyu Park[5], Thomas Randall[7], Amy Ly[8], Cesar M. Castro [1,9] ✉ & Hakho Lee [1,2] ✉

Addressing the global disparity in cancer care necessitates the development of rapid and affordable nucleic acid (NA) testing technologies. This need is particularly critical for cervical cancer, where molecular detection of human papillomavirus (HPV) has emerged as an accurate screening method. However, implementing this transition in low- and middle-income countries has been challenging due to the high costs and centralized facilities required for current NA tests. Here, we present CreDiT (CRISPR Enhanced Digital Testing) for on-site NA detection. The CreDiT platform integrates i) a one-pot CRISPR strategy that simultaneously amplifies both target NAs and analytical signals and ii) a robust fluorescent detection based on digital communication (encoding/ decoding) technology. These features enable a rapid assay (<35 minutes) in a single streamlined workflow. We demonstrate the sensitive detection of cell-derived HPV DNA targets down to single copies and accurate identification of HPV types in clinical cervical brushing specimens ($n = 121$).

The global incidence of human papillomavirus (HPV)-driven cancers, such as cervical, anal, and head and neck cancer, is on the rise, especially in underserved regions with limited healthcare infrastructure. Cervical cancer alone accounts for over 600,000 new cases a year, with more than 80% of them occurring in low- and middle-income countries (LMICs). Prompt and reliable triaging of high-risk HPV (hrHPV) is essential to offset severe pathology bottlenecks in resource-limited regions and circumvent geographical and/or socioeconomic barriers to effective cervical cancer screening[1–3].

Molecular diagnostics for hrHPV have demonstrated superior sensitivity (96–100%) and specificity (90–100%) when compared to standard cytology (e.g., Pap smear)[4]. Consequently, developed countries are increasingly shifting towards centralized nucleic acid (NA) testing as the sole screening method for identifying high-risk cases. However, implementing such tests in LMICs still faces technical challenges. The primary assay method, polymerase chain reaction (PCR), incurs high fixed and operational costs, straining public health budgets[5,6]. Centralized tests are also incompatible with the "screen-and-treat" strategy employed in LMICs—the one-time window of intervention with cervical ablative therapy. Visual inspection with acetic acid (VIA) is a rapid, inexpensive alternative but suffers from poor accuracy (i.e., high rates of false results)[7,8]. These realities underline the unmet need for fast, accurate hrHPV testing, ideally administered at point-of-care (POC) in LMIC settings.

[1]Center for Systems Biology, Massachusetts General Hospital Research Institute, Boston, MA, USA. [2]Department of Radiology, Massachusetts General Hospital, Harvard Medical School, Boston, MA, USA. [3]Bionanotechnology Research Center, Korea Research Institute of Bioscience and Biotechnology (KRIBB), Daejeon, Republic of Korea. [4]Department of Electrical Engineering and Computer Science, Massachusetts Institute of Technology, Cambridge, MA, USA. [5]Department of Chemical and Biomolecular Engineering (BK21 Four), Korea Advanced Institute of Science and Technology (KAIST), 291 Daehak-ro, Yuseong-gu, Daejeon 34141, Republic of Korea. [6]CaNCURE program, College of Science, Northeastern University, Boston, MA, USA. [7]Department of Obstetrics and Gynecology, Massachusetts General Hospital, Boston, MA, USA. [8]Department of Pathology, Massachusetts General Hospital, Harvard Medical School, Boston, MA, USA. [9]Department of Medicine, Massachusetts General Hospital, Harvard Medical School, Boston, MA, USA. [10]These authors contributed equally: Chang Yeol Lee, Hyunho Kim, Ismail Degani. ✉e-mail: Castro.Cesar@mgh.harvard.edu; hlee@mgh.harvard.edu

Clustered regularly interspaced short palindromic repeats (CRISPR) technologies are emerging as a powerful tool in nucleic acid (NA) detection[9–11]. One key advantage of this approach is its capability for sequence-specific signal amplification. When a CRISPR-associated (Cas) protein binds to its target NA through guide RNA (gRNA), the resulting complex becomes an active endonuclease capable of non-selectively degrading single-stranded NAs. This property has been harnessed to amplify analytical signals, with activated Cas proteins cleaving abundant non-targeted signaling probes such as single-stranded NAs tagged with a fluorescent dye (F) and quencher (Q) pair. The rapid and isothermal nature of Cas reactions has also facilitated on-site operations (Supplementary Table 1). We reasoned to adopt this Cas mechanism for fast, accurate NA detection in resource-limited settings.

Here, we report a robust POC technology, termed CreDiT (for CRISPR Enhanced Digital Testing), devised for decentralized HPV testing and HPV-related cancer screening. CreDiT integrates two distinct strategies: (i) a Cas reaction that concurrently amplifies and detects target NAs and (ii) an efficient signal processing inspired by digital radio communications[12,13]. This combination confers notable advantages: (i) CreDiT assays are fast (<20 min), isothermal, and conveniently performed in a single tube; (ii) signal detection is resilient against extraneous noise and interference, enhancing assay reliability and precision; and iii) the detection system is compact for field use without requiring complex optics. Exploiting these advantages, we implement a CreDiT platform positioned for cancer diagnostics. Specifically, we design CreDiT probes for hrHPV genes (HPV16, HPV18, HPV45, HPV31, HPV33, and HPV58) and oncoprotein mRNAs ($E6$, $E7$, and $p16^{INK4a}$), unify a workflow for NA extraction and CreDiT assay, and

build a portable CreDiT system to process multiple (12) samples. The system detects cell-derived HPV DNA targets down to single copies in 35 min (including DNA extraction) and accurately classifies hrHPV types in clinical cervical brushings ($n = 121$). We further expand CreDiT's utility and clinical impact by detecting hrHPV in anal swab samples from separate patients ($n = 48$)[14,15]. The results underscore CreDiT's versatility and potential for HPV-related clinical assessments.

## Results

### CreDiT platform development

We devised the CreDiT assay to minimize user intervention, enabling process automation and effectively achieving a sample-in-answer-out test (Fig. 1a). Our NA assay thus integrated NA amplification and Cas12a-based NA recognition (Fig. 1b). During CreDiT's initial phase, the polymerase enzyme separates the double-stranded target DNA, creating a single-stranded region for the gRNA molecule to recognize. This recognition triggers the Cas12a enzyme within the Cas12a/gRNA complex to act as a nuclease and cleave the F-Q DNA probes, generating a fluorescent signal. However, activated Cas12a can also interfere with RPA by cleaving the target DNA itself (cis-cleavage) and breaking down the RPA primers (trans-cleavage). Reassuringly, our kinetic analysis indicates that RPA is faster than these competing reactions (see Supplementary Note for details). This dominance of RPA ensures sufficient amplification of the target DNA within the CreDiT format.

All assay steps are performed at a fixed temperature (42 °C, isothermal) without requiring external interruptions. Furthermore, the assay is fast and sensitive, as it concurrently amplifies target DNA (RPA) and analytical signals (Cas12a/gRNA). Similar one-pot approaches have

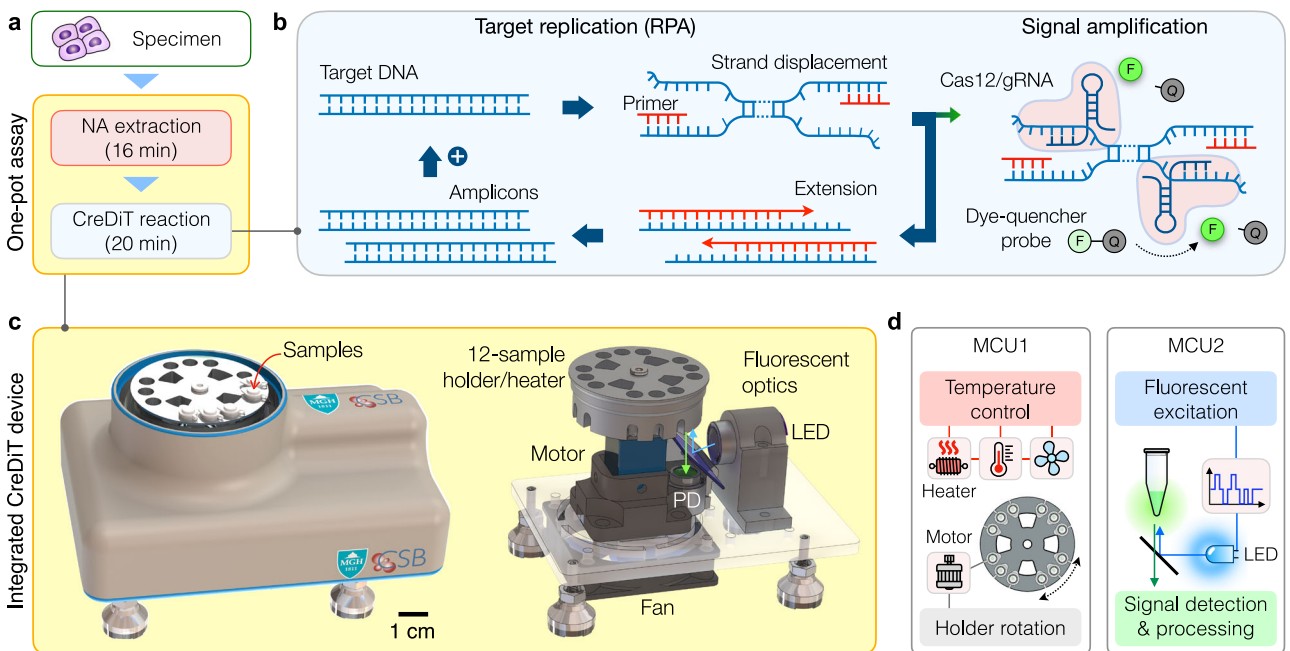

**Fig. 1 | CreDiT assay for HPV detection. a** Streamlined onsite workflow. The assay executes the nucleic acid (NA) extraction and the CreDiT reaction in a single pot, minimizing the hands-on times. The total assay time is about 36 min. **b** CreDiT assay principle. Two reactions concurrently occur: recombinase polymerase amplification (RPA) and Cas12a-based signal generation. RPA is the major reaction that replicates the target DNA. During RPA, the polymerase displaces the DNA duplex, exposing single-stranded regions recognizable by Cas12a/gRNA complexes. This recognition event activates the complexes to cleave dye-quencher (F-Q) probes, generating fluorescent signals. As the RPA cycles proceed, more Cas12a/gRNA complexes are activated, amplifying the overall signal. All reactions are carried out at a constant temperature (42 °C) without requiring external interruptions. **c** A

compact, integrated assay system was developed. The device had a turret-shaped heating block accommodating 12 samples and an optical module for fluorescent detection (LED light-emitting diode, PD photodiode). The heating block, mounted on a motor, rotated to position each sample for fluorescent measurements. The device had a small form factor ($14 \times 11 \times 6.5\ cm^3$) and connected to a computer. Custom-designed software running on a computer presented a graphical user interface for device control and real-time data analysis. **d** Two microcontroller units (MCUs) automated the CreDiT assay processes. MCU1 controlled heating elements to maintain the desired temperatures and rotated the heating block during optical detection. MCU2 generated waveforms for fluorescent excitation and detected the resulting signal.

been described[16,17]; our analysis may be applicable to understand their underlying reaction mechanisms and inform optimizations.

To translate CreDiT's practical advantages into on-site testing, we constructed an automated assay system (Fig. 1c and Supplementary Movie 1). The system hardware integrated a rotating sample holder, heating components, an optical detection module, and microcontroller units (detailed schematic in Supplementary Fig. 1). The sample holder, machined in aluminum, had embedded resistive heaters and temperature sensors and was turret-shaped to accommodate 12 standard PCR tubes (6 mm in diameter). The optical module was configured for fluorescent measurements, including a light-emitting diode (LED) for excitation and a photodiode for detection.

We controlled the system operation using two onboard microcontrollers (Fig. 1d), one for sample heating and the other for signal detection. The heating controller kept the sample holder at a desired temperature for a given assay step (e.g., NA extraction, CreDiT reaction). It executed a feedback loop control, adjusting input powers to heaters and a fan based on the temperature sensor readouts. When the reaction was over, this controller also rotated the sample holder using a stepping motor and positioned each sample for fluorescent detection. The signal detection controller directed optical measurements. It generated waveforms to modulate fluorescent excitation and processed the resulting light signals for demodulation. The approach enabled multi-sample measurement with an identical optical module, ensuring consistent signal detection and simplifying the system design. The developed device was compact with a form factor of $14 \times 11 \times 6.5 \, cm^3$. We also implemented user-interface software for device control and real-time data analysis on a computer (Supplementary Fig. 2).

## Robust optical detection for POC operation

Ideal point-of-care (POC) devices should have fewer components to minimize design complexity and reduce cost, but this requirement may limit system reliability and analytical capacity. Fluorescent devices are in such a category. Simple optical setups may be used, although they are susceptible to environmental and system noises. Conversely, reliable fluorescent measurements use calibrated, high-end systems, challenging POC operations. To address this issue, we designed the CreDiT optical system to generate high-quality signals in a compact device.

We specifically adopted the Walsh–Hadamard transform (WHT)[18], a digital signal processing technique used in wireless communications[19]. The WHT is similar to the Fourier transform (FT) but uses non-sinusoidal, rectangular waves (Walsh functions) as an orthogonal basis (Supplementary Fig. 3a). Any time-domain signals can be decomposed into a weighted sum of Walsh functions[12,20]. Moreover, Walsh functions are uniquely defined by a sequence index, which is analogous to the frequency in FT. With these attributes, the WHT can compress time-domain signals into a simple collection of coefficients for Walsh sequence values (Supplementary Fig. 3b). The WHT method offers further advantages[21]. (i) Walsh functions, inherently binary in nature, are readily compatible with digital electronics. (ii) The signal processing algorithm, referred to as the fast Walsh–Hadamard transform (FWHT), is lightweight and efficient for portable computing, requiring only additions and subtractions of real numbers (Supplementary Fig. 4). This contrasts with the fast Fourier transform (FFT) algorithm, which involves multiplying complex numbers. (iii) More importantly, the WHT is more robust to external interference than other signal processing techniques. The WHT spreads the original signal over a wide frequency band[13]. Such spreading effectively reduces the relative influence of interfering signals, particularly those with narrow bandwidths[22].

To apply the WHT in fluorescent detection, we first generated a signal for fluorescent excitation (Fig. 2a, left). We specified values for the Walsh sequency indices and converted these values into a time-domain waveform using the FWHT. This waveform drove an LED, exciting fluorescent dyes in a sample. In the detection path (Fig. 2a, right), we obtained the emitting light using a photodiode (PD) and decomposed the signal via inverse FWHT. This process revealed the Walsh sequence indices used to modulate the fluorescent excitation, and the peak values at these indices were proportional to the fluorescent intensity (i.e., the CreDiT signal). The measurement was completed in 2.5 ms, minimizing the risk of photobleaching fluorophores.

We determined the optimal Walsh sequence by assessing the system's performance under the excitation of different Walsh waveforms. Specifically, we measured the fluorescent signals from a standard sample ([fluorophore] = 100 μM) while sweeping Walsh sequence (Supplementary Fig. 5a). The system achieved a high signal-to-noise ratio (SNR) within the range of sequences from 97 to 130 (Supplementary Fig. 5b). We selected sequence 107 for its peak SNR in our current prototype. For different systems, the sequence number can be readily adjusted to maximize their SNR.

We further compared the WHT-based detection (CreDiT) with other frequently used techniques: constant illumination and analog lock-in (Fourier) methods. For this task, we programmed the onboard microcontroller to generate either a direct current (DC) or a sinusoidal waveform for LED excitation. Acquired fluorescent signals from a sample ([fluorophore] = 100 μM) were then numerically processed to extract fluorescent intensities (Methods). This approach allowed for an unbiased comparison among signal processing algorithms by keeping the optical setup and electronics identical. The WHT method outperformed the other two, displaying the highest SNR (Fig. 2b). In fluorophore-titration measurements, the WHT detection achieved a superior sensitivity, with its limit of detection (0.4 pM) about 2200-fold lower than that (1 nM) of the lock-in method (Fig. 2c). The results by the WHT detection also showed excellent linear correlation ($R^2 = 0.99$) with a commercial plate reader (Fig. 2d), which confirmed the reliable analytical performance of the developed signal-processing approach.

## Temperature control for CreDiT reaction

To enable consistent CreDiT reactions, we engineered precise temperature control for the assay device. We first optimized the sample holder design to facilitate rapid and even heating for all 12 samples. Through numerical simulations, we determined that a holder featuring hollow pockets would heat up more rapidly than a solid one, primarily due to the reduced thermal mass (Supplementary Fig. 6). Based on this design insight, we proceeded to fabricate a holder in aluminum. The holder had four quadrants (Fig. 2e); each quadrant accommodated three samples and was embedded with a pair of a heater and a temperature sensor. To regulate the holder temperature, we programmed a microcontroller for feedback control: it read temperature sensors every 100 msec, averaged their outputs, and generated a control signal for heating elements or a cooling fan. Applying the method, we could create reliable temperature profiles suited for different assay steps (Fig. 2f). On average, the accuracy was within ±0.3 °C of the target temperature, and the precision was 0.6% as measured by the coefficient of variation (Fig. 2g). The temperature was also uniform across the 12 samples as evidenced by the overall heat distribution recorded throughout the assay (Supplementary Movie 1).

## Assay optimization for POC operation

We proceeded to fine-tune the entire CreDiT assay process, aiming to establish a streamlined workflow. For NA extraction from cells, we chose to apply enzymatic cell digestion, a method that does not require external instruments (e.g., centrifuge and spin-columns) and can be controlled by adjusting the temperature (Fig. 3a). We used a lysis buffer containing proteinase K (Pro K) to release cellular NAs and protect them from nucleases. During the lysis, the sample was heated to 56 °C to enhance Pro K activity. Using cervical cancer cells (Ca Ski),

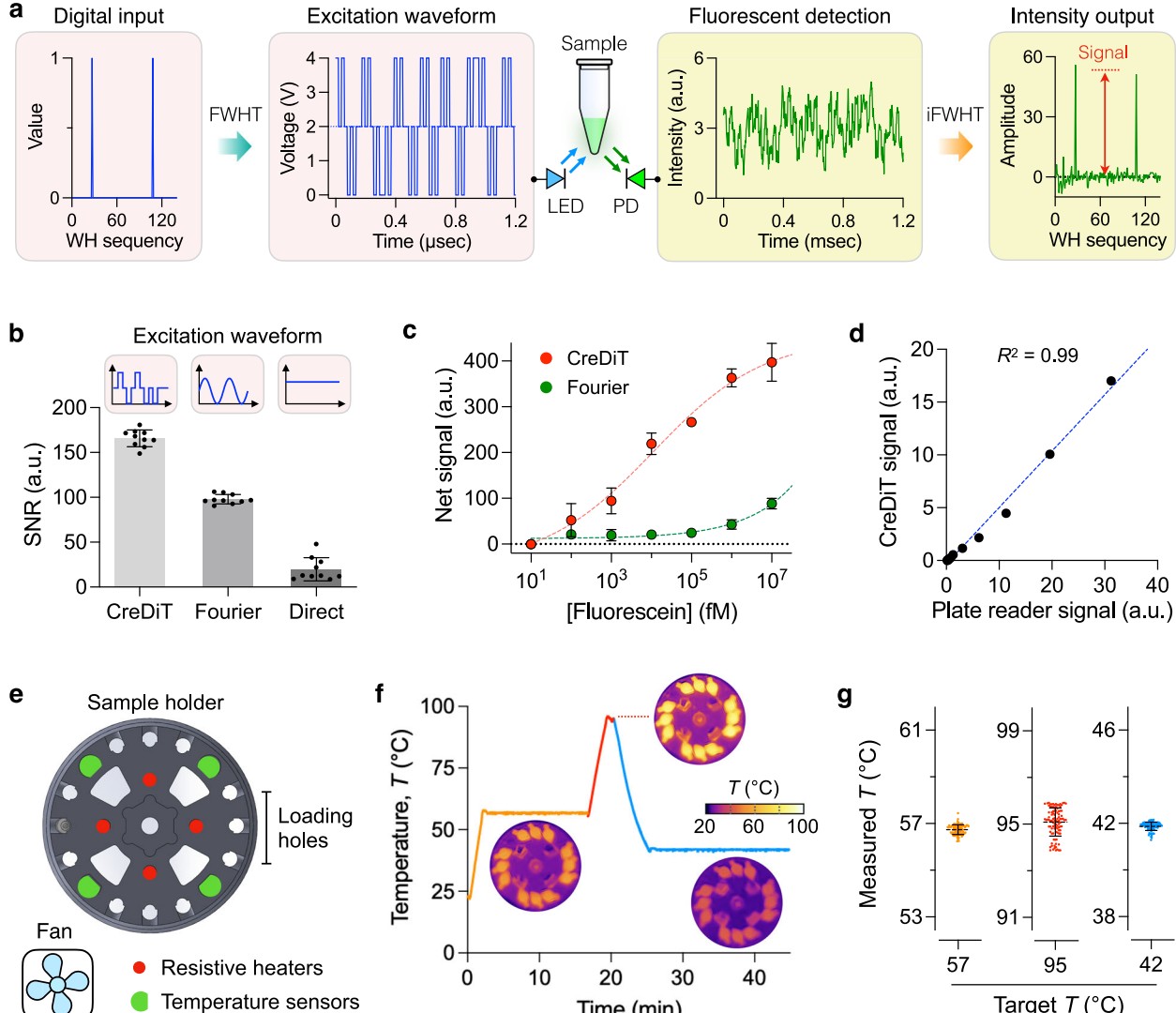

**Fig. 2 | CreDiT system engineering. a** CreDiT approach to fluorescent detection. The digital input, represented by a Walsh–Hadamard (WH) sequence, is transformed into a fluorescent excitation waveform using a fast WH transform (FWHT). The resulting signal from a sample is processed via inverse FWHT (iFWHT) to recover WH sequence, whose amplitude is proportional to the fluorescent intensity. **b** Digital CreDiT detection outperformed analog lock-in (Fourier) and direct fluorescent detection, exhibiting a higher signal-to-noise ratio (SNR) than the others. Data were displayed as mean intensity ± s.d. from technical replicates ($n = 10$). **c** Samples with varying fluorescein concentrations were measured. With its high SNR, digital CreDiT detection achieved >2200-fold lower detection limit (0.4 pM) than the analog lock-in method (Fourier; 1 nM). Data were displayed as mean ± s.d. from triplicate measurements. **d** The digital CreDiT results from (**c**) also showed an excellent correlation ($R^2 = 0.99$) with those measured by a conventional plate reader. **e** Sample heating setup. The sample holder accommodated 12 samples for heating and was embedded with four pairs of a heater and a temperature sensor. An external fan was used to expedite sample cooling. **f** Sample temperature profile. The system reached different temperature targets for the CreDiT assay (e.g., NA extraction, CreDiT reaction). Thermal imaging (inset) confirmed uniform heating of all 12 samples. **g** The heating system maintained the target temperature with relative variations <0.6%. Data were shown as mean ± s.d. from repeated temperature measurements ($n = 100$). a.u. arbitrary unit. Source data are provided as a Source Data file.

we optimized the lysis time (15 min) and the buffer composition ([Pro K] = 0.5 mg/mL) for the maximal NA yield (Fig. 3b). Following the NA extraction, we deactivated Pro K by elevating the buffer temperature; this process was necessary to prevent Pro K from affecting the downstream CreDiT reaction. We found that a brief incubation (1 min) at 95 °C was sufficient (Supplementary Fig. 7).

The performance of our extraction protocol was similar to that of a standard method (commercial spin-column). For this evaluation, we extracted NAs from cells (Ca Ski) and subjected them to PCR and CreDiT analysis. NAs from both extraction methods had similar quality (Supplementary Fig. 8a) and generated strong PCR bands of comparable size and intensity (Fig. 3c). The extraction yields were also comparable between the two extraction methods, as confirmed by PCR and CreDiT assays producing statistically indistinguishable (unpaired, two-

sided *t*-test) signals (Fig. 3d). When compared to the method using only thermal lysis (95 °C) for NA release, the CreDiT extraction yielded six times more NA (Supplementary Fig. 8b); this improvement was likely due to the additional benefits of Pro K, notably the degradation of nucleases (i.e., RNases and DNases) that break down NA.

We further assessed the feasibility of storing samples and reagents under ambient conditions; this capability is desirable for promoting CreDiT applications in LMIC sites. From the perspective of sample storage, we tested the stability of cells in our extraction buffer. We spiked cells (Ca Ski) in the buffer and stored sample aliquots either in a refrigerator (4 °C) or under ambient conditions before NA extraction. For both storage conditions, we observed no notable decrease in NA amounts for at least two weeks (Supplementary Fig. 9). Regarding CreDiT reagents, we premixed the components (RPA

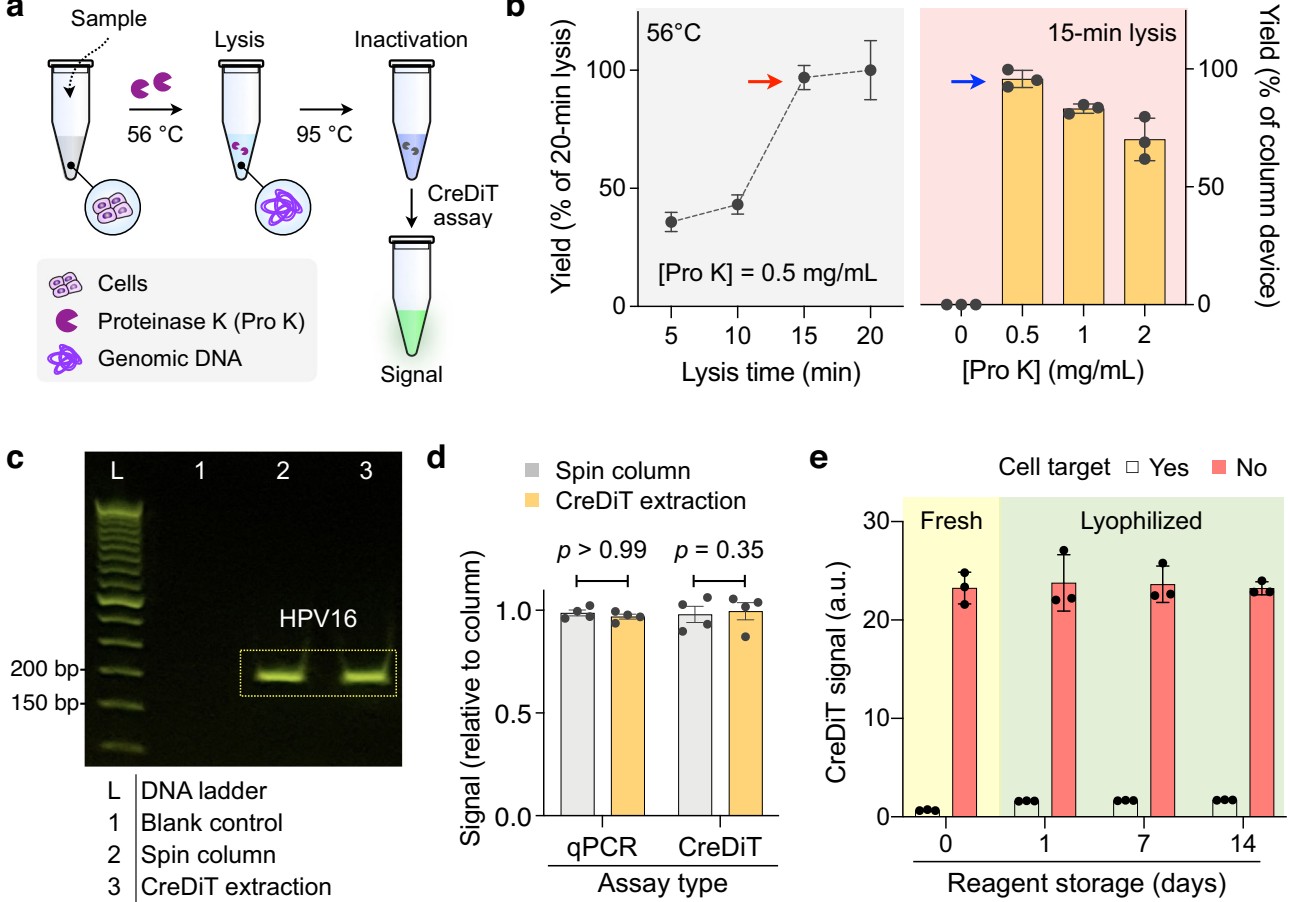

**Fig. 3 | Assay optimization for POC applications. a** NA extraction protocol. Cells are lysed by proteinase K (Pro K) to release NAs. Subsequently, Pro K is heat-deactivated (95 °C), and the CreDiT reaction is initiated. **b** The lysis time and Pro K concentration ([Pro K]) were optimized to maximize the NA extraction yield. Samples containing Ca Ski cells ($2.5 \times 10^4$ cells/mL) were analyzed for HPV16. (Left panel) The extraction yields reached a plateau (red arrow) within 15-min of lysis. HPV16 signals from 20-min lysis were used for normalization. (Right panel) The extraction yield was maximal (blue arrow) at [Pro K] = 0.5 mg/mL and comparable to that of a spin column (gold standard). Data were displayed as mean ± s.d. from technical triplicates. **c** Extracted NAs were PCR-amplified for the HPV16 gene. Gel electrophoresis showed comparable bands between the CreDiT and the spin-column extraction methods. bp, base pairs. **d** Both extraction methods (CreDiT, spin column) yielded statistically identical results (unpaired, two-sided *t*-test) in analytical assays. Data were displayed as mean ± s.d. from technical quadruplicates. **e** CreDiT reagents were lyophilized and stored under ambient conditions. The lyophilized reagents demonstrated consistent activity for at least two weeks. Ca Ski cells ($2.5 \times 10^4$ cells/mL) were analyzed for HPV16. Data were displayed as mean ± s.d. from technical triplicates. a.u. arbitrary unit. Source data are provided as a Source Data file.

primers, reporter probe, Cas12a gRNAs, and Cas12a protein) and lyophilized the mixture (see Methods) to facilitate its transport and storage. When evaluated via the CreDiT assay, the lyophilized premix retained its efficacy for at least two weeks in storage at ambient conditions (Fig. 3e). Overall, these features could mitigate logistic challenges and promote CreDiT usability in resource-limited settings even without cold chain solutions.

## CreDiT probe design and validation

We prepared a library of CreDiT probes for hrHPV detection. For a given target DNA, CreDiT used a pair of probes, one forward and one reverse, to span both ends of the target sequence (Fig. 4a). At each end of the sequence, we designed an RPA primer and a Cas12a gRNA against a target region. The primer, typically 30–33 nucleotides in length, served to amplify the target region, and the gRNA featured a spacer sequence allowing Cas12a to bind to a region positioned a few bases (2–10 nucleotides) downstream from the primer binding site (detailed design flow in Supplementary Fig. 10). Because Cas12a was activated by a single-strand DNA, the gRNAs can be designed without the constraint of a protospacer adjacent motif (PAM) sequence, offering enhanced flexibility. Capitalizing on this

flexibility, we designed CreDiT probes for a diverse range of targets (see Supplementary Tables 2, 3): (i) genomic DNA of hrHPV subtypes prevalent in Africa (HPV16, HPV18, HPV31, HPV33, HPV45, and HPV58); (ii) oncogenic *E6/E7* mRNA whose presence may indicate progressive HPV16 infections; and (iii) *p16^INK4a* (*CDKN2A*) mRNA that is upregulated in transformed cells as a consequence of *E6/E7* expression.

We validated the designed probes in the CreDiT assay. We first varied the composition of the assay reagents and monitored the resulting CreDiT signals (Fig. 4b and Supplementary Fig. 11). As expected, the fluorescent signal was only observed when all CreDiT reagents and the target DNA were present. We further observed the synergistic benefit of recognizing both the forward and reverse regions in a target DNA. The combination of forward and reverse gRNAs produced stronger analytical signals than individual gRNAs (Supplementary Fig. 12). We also designed an additional gRNA that recognized a middle region of the target DNA (Supplementary Table 2). However, this gRNA displayed the weakest signal generation, and incorporating it into the assay lowered the overall signal, likely due to competition with the more efficient forward and reverse gRNAs for Cas12a binding.

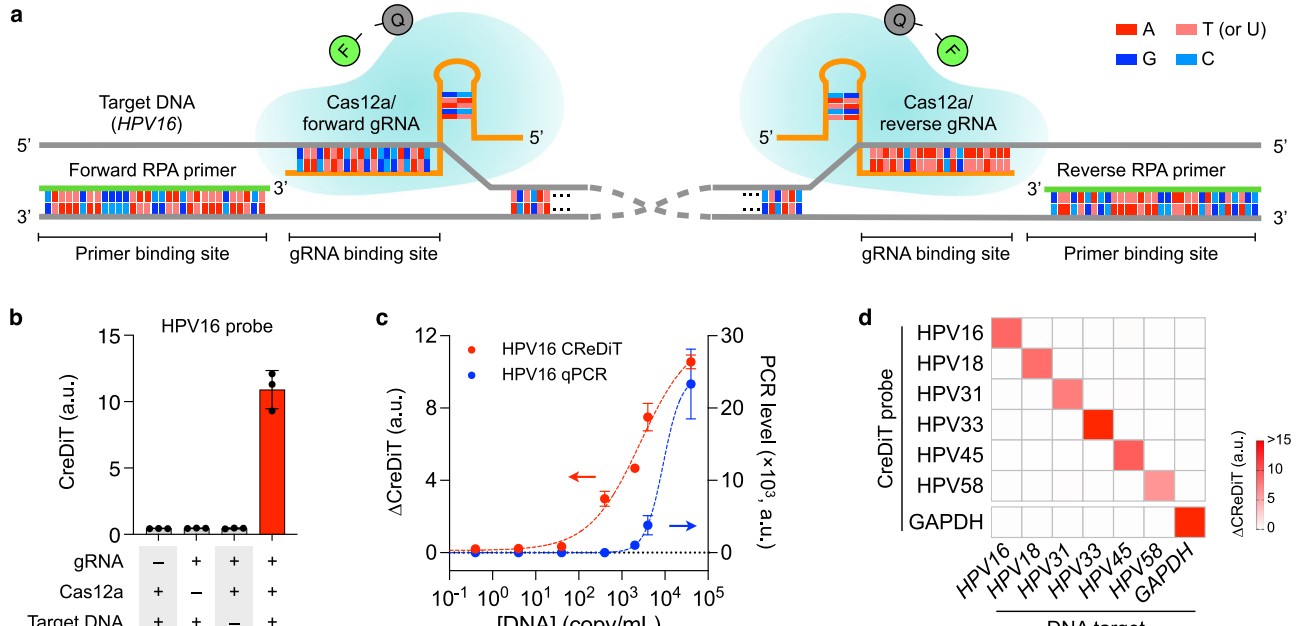

**Fig. 4 | CreDiT probe design. a** For a given target DNA (gray backbone), a pair of CreDiT probes (forward and reverse) were designed, with each probe consisting of an RPA primer (green backbone) and a Cas12a/gRNA complex (orange backbone). Upon binding to the target DNA, the primer exposes a downstream sequence complementary to gRNA. The Cas12a/gRNA complex then recognizes its target and becomes an active endonuclease to cleave F-Q reporters. HPV16 probes are shown. **b** Assay validation with varying reaction conditions. HPV16 DNA was used as a model target ([DNA] = 4 × 10⁷ copies/mL). A high signal was only observed when all assay components were present. Data were displayed as mean ± s.d. from technical triplicates. **c** CreDiT (20 min) and qPCR (1 h) assays were used to analyze samples with different target DNA concentrations. The CreDiT assay exhibited a lower detection limit (40 copies/mL) and a wider dynamic range (4 orders of magnitude) than qPCR (detection limit, 400 copies/mL; dynamic range, 2.5 orders of magnitude). ΔCreDiT is the background-subtracted signal. Data were displayed as mean ± s.d. from technical triplicates. **d** Probe selectivity. A panel of probes was designed for hrHPV targets (HPV16, HPV18, HPV31, HPV33, HPV45, and HPV58) and *GAPDH*. These probes detected their intended targets with negligible off-target signals. [Target DNA] = 4 × 10⁷ copies/mL. The heatmap displays mean values from technical triplicates. a.u. arbitrary unit. Source data are provided as a Source Data file.

Next, we compared CreDiT's performance with the established gold standard, qPCR, and a two-step assay, where RPA (20 min) and Cas12a detection (30 min) were carried out sequentially. We used synthetic HPV16 DNA as a model target. CreDiT demonstrated superior sensitivity compared to qPCR (Fig. 4c) and the two-step assay (Supplementary Fig. 13). CreDiT's limit of detection (LOD) was down to 40 copies/mL, whereas LODs for qPCR and the two-step assay were around 400 copies/mL. CreDiT also offered an advantage in assay time, requiring only 20 min compared to 1 h for qPCR and 50 min for the two-step assay. Furthermore, the designed CreDiT probes exhibited excellent selectivity with minimal cross-reactivity, as confirmed by the CreDiT assay (Fig. 4d and Supplementary Fig. 14) and gel electrophoresis (Supplementary Fig. 15).

## CreDiT assay characterization

With the detection system and probes ready, we characterized the entire CreDiT assay flow, beginning with NA extraction from cervical cancer cells. To assess the assay kinetics, we extracted DNA from varying numbers of Ca Ski cells (HPV16+) and performed CreDiT assays, monitoring the signal evolution over time. Even at low cell concentrations, the signal rose rapidly above the background (Fig. 5a); this can be attributed to the dual enzymatic activities in CreDiT, namely, RPA for NA duplication and Cas12a reaction for signal amplification. From the kinetics data, we observed that the limit of quantification (LOQ) reached below a single cell cutoff (in 25 μL reaction volume) in about 20 min (Fig. 5b); we thus set the detection time to 20 min. In cell-titration experiments (Fig. 5c), the 20-min CreDiT assay achieved the LOQ of 0.8 cells (33 cells/mL) and a dynamic range spanning five orders of magnitude.

We expanded our evaluation to other hrHPV probes, using a panel of cervical cancer cell lines of different HPV subtypes: Ca Ski (HPV16+), SiHa (HPV16+), HeLa (HPV18+), and C-33A (HPV−). DNAs were extracted from these cells (~10⁵ cells) and subjected to multi-sample CreDiT assays. We used the *GAPDH* signal as a positive control to confirm the presence of sufficient cells. The hrHPV profiling results matched with known cellular genotypes[23,24]. Only on-target signals were positive, while off-target signals were all close to the background (Fig. 5d).

We further applied CreDiT to detect mRNA targets (*E6*, *E7*, and *p16^INK4a*), which are overexpressed in invasive, high-grade dysplasia, a precursor to malignant cervical cancer[25]. Detecting these targets can help identify high-risk cases, minimizing overtreatment. To adapt CreDiT for mRNA assays, we simply introduced reverse transcriptase into the CreDiT reagent mix, which enabled the reverse transcription to proceed concurrently with the CreDiT reaction (see Methods). The CreDiT results (Fig. 5e) were consistent with those from gold standard RT-PCR. They showed a qualitative match with PCR-product electrophoresis (Fig. 5f and Supplementary Fig. 16) and a good correlation (*r* = 0.88) with quantitative PCR results (Supplementary Fig. 17).

## CreDiT clinical testing

We finally applied CreDiT to analyze human clinical samples. We used cervical or vaginal brushing specimens (*n* = 121) collected during routine gynecologic evaluation (see Table 1). To encompass diverse demographics, we collected liquid-based samples from patients at a tertiary-care US hospital (Massachusetts General Hospital) as well as its affiliated primary care clinics serving multiple low-resource neighborhoods across the city (Boston, Massachusetts, USA) and adjacent areas. The samples were then assessed for HPV status in a clinical

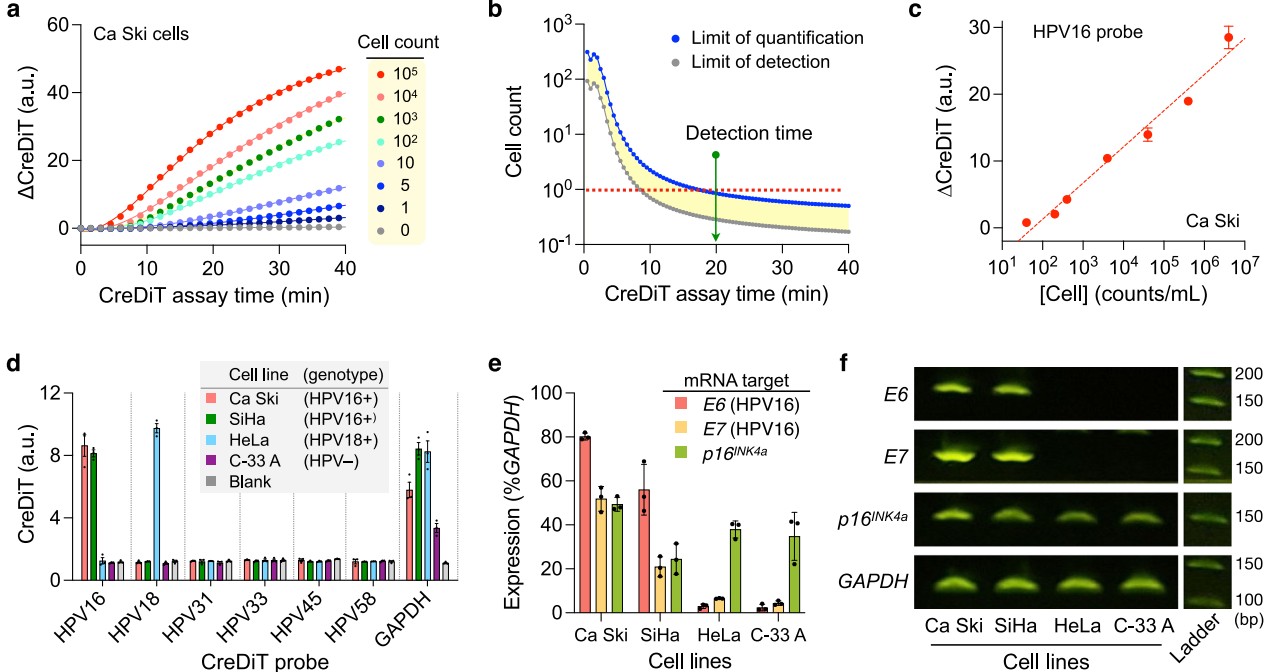

**Fig. 5 | CreDiT assay characterization with cellular samples. a** Temporal evolution of CreDiT signal. A varying number of Ca Ski cells were analyzed for HPV16. The signal started rising above the background about 10 min after the CreDiT reaction started. ΔCreDiT is the net signal with background subtracted. **b** Limits of detection (LOD) and quantification (LOQ) were estimated at different assay times from (**a**). The LOQ reached below a single cell cutoff (in 25 μL reaction volume) after 20 min of the assay start. The detection time was set to this moment. **c** The 20-min CreDiT assay achieved an LOQ of 0.8 cells (33 cells/mL) and a dynamic range over five orders of magnitude. Data were displayed as mean ± s.d. from technical triplicates. **d** A panel of CreDiT probes was tested on cervical cancer cells (1000 cells per sample). The CreDiT assay accurately detected HPV genotypes in different cervical cancer cells. The GAPDH probe was used to ascertain cellularity in samples. Data were displayed as mean ± s.d. from technical triplicates. **e** The CreDiT assay was compatible with mRNA detection. CreDiT probes were designed for mRNA counterparts of *E6* and *E7* oncoproteins of HPV16 and *p16^INK4a*. The assay detected high expression of *E6* and *E7* in HPV16-positive cells (Ca Ski, SiHa). The expression of *p16^INK4a* was elevated in all four cell lines. Samples contained about 1000 cells. Data were displayed as mean ± s.d. from technical triplicates. **f** The CreDiT results in (**e**) qualitatively matched gel electrophoresis results of RT-PCR products of the same mRNA targets. a.u. arbitrary unit, bp base pairs. Source data are provided as a Source Data file.

pathology laboratory (BD Onclarity HPV Assay system). These qPCR-based results served as our gold standard (see Supplementary Data 1 for the results). Residual aliquots of brushing specimens were subjected to CreDiT, including NA extraction and target DNA detection (HPV16, HPV18, HPV31, HPV33, HPV45, HPV58, and *GAPDH*). All samples generated high *GAPDH* signals, passing the cellularity check (Supplementary Fig. 18).

**Table 1 | Demographic and clinical information of the cervical brushing samples**

| Cervical brushing Case | Total | | |
|---|---|---|---|
| | Negative | HPV | |
| | 52 | 69 | 121 |
| Age | | | |
| Median | 41 | 42 | 41 |
| Range | 23-69 | 21-87 | 21-87 |
| Gender | | | |
| Male | 0 | 0 | 0 |
| Female | 52 (100%) | 69 (100%) | 121 (100%) |
| Subtype | | | |
| HPV16 | - | 36 (53%) | 38 (30%) |
| HPV18 | - | 15 (22%) | 16 (12%) |
| HPV45 | - | 20 (29%) | 21 (17%) |
| Others† | - | 14 (20%) | 15 (12%) |

†This category combines hrHPV subtypes of HPV31, 33, 35, 39, 51, 52, 56, 58, 59, 66, and 68.

Figure 6a displays the HPV-CreDiT profile of the brushing samples and their pathology diagnoses. CreDiT accurately distinguished HPV-positive from negative samples and identified three major hrHPV types (HPV16, HPV18, and HPV45), agreeing with clinical diagnostics. For other HPV targets (HPV31, HPV33, and HPV58), CreDiT results were a subset of the clinical "others" category; the clinical diagnostics grouped ten targets (HPV31, HPV33, HPV35, HPV39, HPV51, HPV52, HPV56, HPV58, HPV66, and HPV68) in a single class.

For three major hrHPV targets (HPV16, HPV18, HPV45), we further assessed CreDiT's diagnostic performance. For a given HPV target, we dichotomized samples according to the clinical HPV status (Fig. 6b). As a diagnostic metric, we used the marker signal scaled to *GAPDH* (e.g., ΔCreDiT$_{HPV16}$/ΔCreDiT$_{GAPDH}$ for HPV16), accounting for the varying cellularity in the samples. Using this metric, we constructed a receiver operating characteristic (ROC) curve (Fig. 6c) and then determined the cutoff for positivity that maximized sensitivity and specificity. With these cutoffs applied, the observed diagnostic accuracies were 0.975 (HPV16), 0.992 (HPV18), and 0.967 (HPV45; Supplementary Table 4).

To demonstrate CreDiT's extended clinical utility in other cancer screening efforts, we analyzed anal Pap test samples in a similar manner (*n* = 48; Supplementary Table 5). HPV infection accounts for 90% of anal cancer cases[26], with HPV16 being the most prevalent[27]. The CreDiT results (Fig. 6d) again showed excellent agreement with clinical diagnostics, achieving high diagnostic accuracy (Fig. 6e).

## Discussion
We have developed and validated CreDiT as a robust and affordable diagnostic technology for NA detection. The assay had excellent

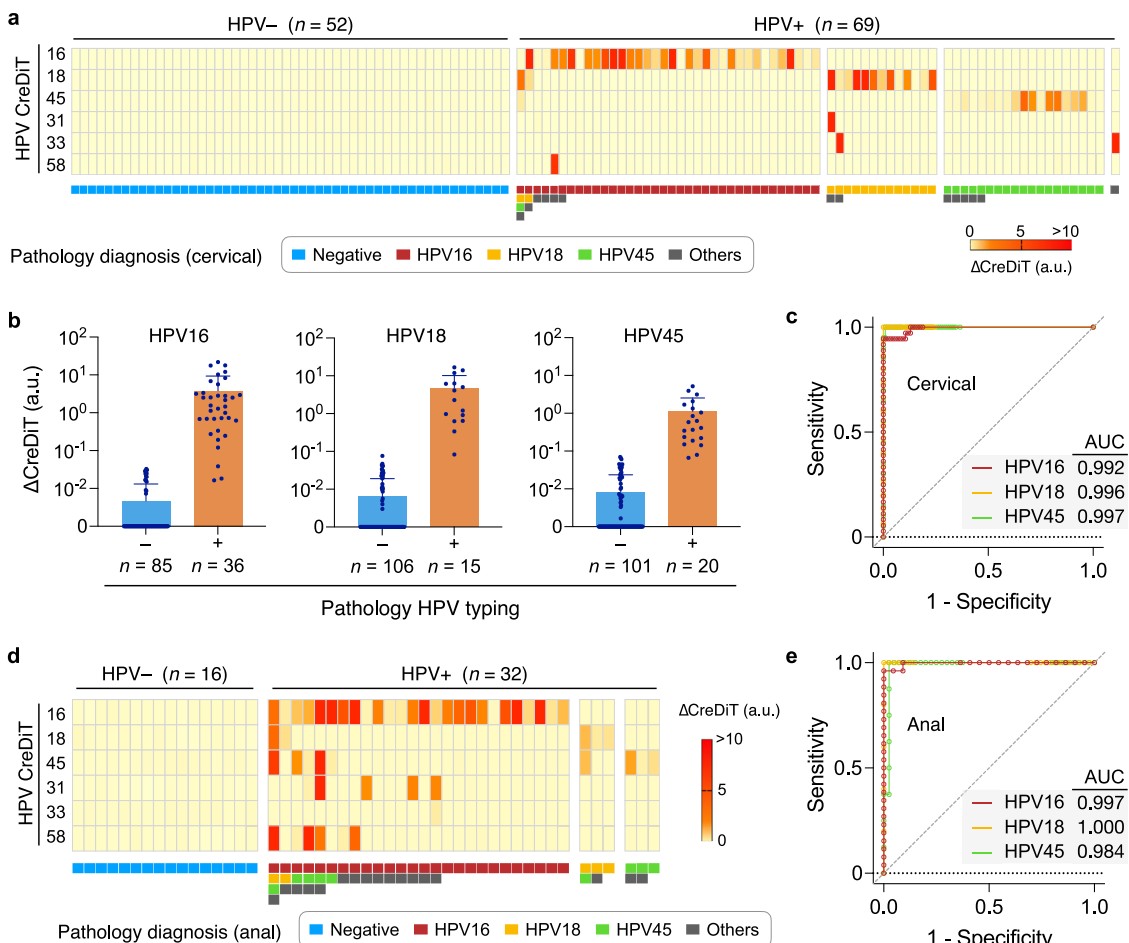

**Fig. 6 | Analysis of clinical samples. a** Cervical brushing samples ($n = 121$) were analyzed via CreDiT. When compared with clinical diagnoses (bottom row), the CreDiT results showed a good concordance. The CreDiT heatmap displays mean values from triplicate measurements. ΔCreDiT is a background-subtracted signal. **b** Samples were categorized according to the clinical diagnoses of three major hrHPV targets (HPV16, HPV18, and HPV45). CreDiT signals were significantly higher in clinically positive samples. The $p$ values were <$10^{-6}$ for all three targets from the unpaired two-sided $t$-test. Data were displayed as mean ± s.d. **c** Receiver operating characteristic (ROC) curves were generated for HPV16, HPV18, and HPV45. The predictor was a marker-specific CreDiT signal scaled to the *GAPDH* signal, considering varying cellularity in a sample. Overall, CreDiT achieved high diagnostic accuracy, with the area under the curve (AUC) values >0.99 for all three HPV types. **d** CreDiT was used to analyze anal swab samples ($n = 48$) for anal cancer screening. The results matched the clinical diagnoses (bottom row). The CreDiT heatmap displays mean values from triplicate measurements. **e** ROC curves were constructed using the *GAPDH*-scaled marker expression. The AUC values were >0.98. a.u. arbitrary unit. Source data are provided as a Source Data file.

analytical sensitivity by coupling DNA replication (i.e., RPA) with self-signal amplification of CRISPR/Cas (i.e., collateral cleavage of reporter probes). CreDiT also achieves high specificity; the analytical signal is generated when RPA primers and CRISPR/Cas recognize target sequences. The integrated assay system performed multiple functions following single sample loading, including NA extraction, NA amplification, and optical detection. This system-level engineering markedly simplified the assay procedure while improving overall assay speed (35 min) and minimizing manual sample handling (<3 min hands-on time; see Supplementary Fig. 19 for the detailed assay flow). Furthermore, the system was capable of up to 12 parallel measurements, improving the assay throughput. These technical strengths (sensitivity, throughput, integration, and ease of use) may differentiate CreDiT from other Cas-based point-of-care systems that require microfluidic components[28–30], rely on microscopy[31], or produce semi-quantitative results[32] (see Supplementary Table 1 for comparison). We envision CreDiT to empower LMIC providers with low-cost, decentralized, automated, and rapid readouts to guide "screen and treat" efforts. The same visit turnaround and fractional costs compared to local gold standards will also increase CreDiT's prospects for adoption into clinical workflows.

Molecular HPV testing has emerged as a superior alternative to conventional Pap smear and VIA tests for cervical cancer screening. It offers enhanced convenience and improved detection of high-risk precancerous lesions, particularly cervical intraepithelial neoplasia grade 3 (CIN3). This capability enables timely intervention to prevent the progression of cervical cancer[33–35]. As a primary screening tool, molecular HPV testing further provides a wide range of services, including (i) patient triaging and risk stratification of cytological abnormalities based on HPV subtypes, (ii) follow-up monitoring before or after treatment of CIN, and (iii) surveillance of the HPV epidemic at regional or country levels[36–38]. Encouraged by these clinical benefits, developed countries are increasingly shifting towards centralized HPV DNA testing as the sole screening method for identifying high-risk cases. Unfortunately, similar implementation has been challenged in LMICs due to limited access to test equipment and cost, incurring poor clinical outcomes. For instance, cervical cancer remains the second leading cause of cancer-related deaths in African women. We developed CreDiT to address such disparity by increasing access to molecular tests.

Several technical features elevated CreDiT's practical utility for onsite HPV screening. First, we seamlessly combined NA duplication

(RPA) and signal amplification (Cas reaction) into a unified assay scheme (Fig. 1). This strategy allowed CreDiT to achieve higher (~100-fold) sensitivity than conventional PCR while conveniently carrying out the reaction in a single tube and at a constant temperature (42 °C). Second, we enhanced flexibility in the probe design. CreDiT eliminates the requirement for a PAM sequence in the target DNA; Cas12a can still be activated with crRNA binding to the target DNA's single-strand segment displaced during the RPA reaction (Fig. 4). This merit facilitated the creation of a probe library capable of detecting a wide range of hrHPV targets. Third, we implemented a signal detection algorithm akin to those employed in wireless communication (Fig. 2). Specifically, the algorithm utilizes digitally encoded light to excite samples and then numerically decodes measured fluorescent signals to recover the original waveform. This digital approach offers superior computational efficiency and signal-to-noise ratio compared to conventional analog methods, enabling us to develop a compact CreDiT device suitable for POC applications.

We propose extending the current study to enhance CreDiT's analytical capability and accelerate its clinical translation. An immediate task is to expand CreDiT probes to detect a broader array of hrHPV targets, encompassing HPV35, HPV39, HPV51, HPV52, HPV56, HPV59, HPV66, and HPV68. Such expansion will yield comprehensive insights into the overall prevalence of distinct subtypes, which would be instrumental in devising region-specific or racially tailored countermeasures to combat HPV-associated health challenges (Supplementary Table 3)[3,39,40]. Another improvement would be equipping the assay to detect multiple molecular targets within a single reaction tube. Such multiplexing will adopt orthogonal Cas proteins (e.g., LwaCas13a, LbaCas13a, and PsmCas13b), with each protein type, upon target recognition, selectively cleaving reporter probes of distinct fluorescent colors[41,42]. In signal detection, we could explore applying our digital algorithm for multiplexing. For instance, each fluorescent-detection channel will use its own WH code to generate distinct excitation and response patterns. The total fluorescent signal is then acquired by a single photodetector and numerically decoded into individual channels. This approach would enable multi-fluorescent measurements with minimal optical components (i.e., a single fluorescent detector), substantially simplifying the detection system.

We could further streamline the assay flow. In its current format, CreDiT requires a manual step to add the CreDiT mix into a sample after NA extraction. This step can be simplified with a new sample tube design. This tube would feature two compartments separated by a breakable seal. The top compartment will be pre-filled with CreDiT reagents, while the bottom will house the NA extraction buffer. This dual-chamber design will protect the CreDiT reagents throughout the NA extraction process. Following NA extraction, the CreDiT mix can be effortlessly introduced into the reaction chamber by breaking the seal —an action readily amenable to automation or manual operation.

Finally, we need to deploy CreDiT to LMICs for evaluation by local end users. This real-world assessment will help identify potential oversights or challenges within the overall CreDiT workflow, encompassing logistics, pre-analytical handling, and readouts. The deployment will also foster collaboration with LMIC partners to refine the platform, ensuring its adaptability to specific clinical, economic, and environmental factors. We indeed plan to conduct field trials of CreDiT prototypes in Uganda and Ghana; the insights from these trials will guide further improvements in CreDiT's field performance. Through these advancements, CreDiT will serve as a practical, field-deployable HPV testing solution, enabling the implementation of state-of-the-art HPV guidelines in regions where the impact is the most significant.

## Methods

### Ethical statement

The research protocol received approval from the Partners Healthcare Institutional Review Board (IRB protocol 2022P002938). The investigators adhered to all relevant ethical regulations and institutional guidelines throughout the study. Participants provided written informed consent for the reporting and sharing of individual-level data. The gender of participants was determined based on self-reported information provided on the study consent form. No financial compensation was provided to study participants.

### Materials

We had all oligonucleotides used in this study synthesized by Integrated DNA Technologies, Inc. (IDT; USA). The sequences of the oligonucleotides are listed in Supplementary Table 2. TwistAmp® Basic kit and RevertAid reverse transcriptase (RTase) were purchased from TwistDx™ and Thermo Fisher Scientific (USA), respectively. EnGen® Lba Cas12a (Cas12a), RNase inhibitor (Murine), Pro K, and Luna Universal qPCR Master Mix were purchased from New England BioLabs (USA). All other chemicals were of analytical grade and used without further purification.

### CreDiT device construction

We designed the CreDiT device using computer-aided design software (Solidworks, 2019) and fabricated the main device body through 3D printing (Form2, Formlabs). The device housed a rotating sample holder, a heating unit, an optical module for fluorescent detection, and two onboard computers. (i) The sample holder was cut in a metal block (aluminum 6061-T651, Protolabs) via a computer numerical control (CNC) machine. Into the holder, we inserted four ceramic heaters (HT15W, Thorlabs) for heating and four thermistors (10 kΩ) for temperature reading. The holder was mounted on a stepping motor (NEMA-8, Adafruit) to position each sample for optical detection. We also installed a 60 mm fan (OD6010-12HB, Orion Fans) underneath the holder for cooling purposes. (ii) The optical module had a pair of light a source and a fluorescent detector. The light source consisted of a 470-nm LED (SSL80, OSRAM), an aspheric condenser lens (ACL12708U, Thorlabs), and biconvex focusing lenses (LB1092-A, Thorlabs). The LED was attached to a metal heat sink and driven by a single-supply, low-noise LED current source driver (IRLMS2002TRPBF, Infineon). The fluorescent detector consisted of a filter set (GFP/FITC/Cy2 filter cube, Nikon), a convex lens (LB1761-A, Thorlabs), and a photodiode (S1223, Hamamatsu). The signal from the photodiode was amplified by an operational amplifier with a −3 dB bandwidth of 17 MHz (Analog Devices AD823AN) and digitized by a microcontroller.

### CreDiT device control

We used two onboard computers for the system control. The master unit (Arduino Uno) executed a feedback loop to achieve the desired temperature profile in the sample holder, controlled the stepping motor, and communicated with an external computer. The signal-processing unit (ARM Cortex M4 microcontroller; MK20DX256) generated the waveform LED excitation, digitized the signal from the photodiode, and performed all necessary signal processing, including the Walsh−Hadamard transform. (iv) As a graphical user interface (GUI), we programmed a real-time dashboard using the Qt GUI framework (Supplementary Fig. 2), which communicated with the master unit over a universal serial bus (USB) connection. The firmware of the master unit was written in C++.

### CreDiT signal processing for fluorescent measurements

For fluorescent measurements, we used Walsh functions of size 512. To generate the excitation waveform $E(t)$, we specified an input vector $D$ of length 512. The $i$th element of $D$ represented the value for the $i$th sequence in the WH domain. Applying the Fast Walsh−Hadamard transform (FWHT) to $D$ yielded the waveform $E(t)$. We adjusted the excitation waveform by scaling it and adding a positive voltage offset to drive an LED in the full voltage swing (0−6 V). The resulting input voltage was $E(t) = 3 \cdot W(t) + 3$ (V), where $W(t)$ is the Walsh function. We

subsequently measured a Walsh-modulated fluorescence signal using a photodiode. This signal $S(t)$ was directly digitized (200 kHz) by a 12-bit analog-to-digital converter built into the MCU (MK20DX256). The MCU then performed the inverse FWHT, converting $S(t)$ into the sequency vector $T$. We finally obtained the signal intensity by taking an inner vector product of $D \cdot T$. In the current CreDiT prototype, we used an input vector $D_{107}$, whose 107th element was the only non-zero element. This sequency was close to the operating frequency of 20.9 kHz. A similar approach was applied for the Fourier modulation, with the input voltage $E(t) = 3 \cdot \sin(2\pi f \cdot t) + 3$ (V), where $f$ is the waveform frequency. The average input power to LED had a 4:3 ratio between Walsh and Fourier modulations.

## Cell culture and NA extraction

We purchased Ca Ski (CRL-1550), SiHa (HTB-35), HeLa (CCL2), and C-33A (HTB-31) cells from the American Type Culture Collection and grew them (37 °C, 5% $CO_2$) in the vendor-recommended growth media (RPMI-1640 for Ca Ski and DMEM for SiHa, HeLa, and C-33A) supplemented with 10% fetal bovine serum, 100 U/mL penicillin, and 100 μg/mL streptomycin. All cell lines were regularly tested for mycoplasma contamination using a mycoplasma detection kit (MycoAlert™, Lonza). We extracted NA from these cells following the Pro K-based CreDiT extraction protocol. The lysis buffer had 0.5 mg/mL Pro K in 1× Tris-EDTA (TE; pH 8.0). As a gold standard for NA extraction, we purchased Quick-DNA Miniprep (Zymo Research) and used it according to the manufacturer's protocol.

## CreDiT assay

We prepared the CreDiT assay mix (21.75 μL) by combining target-specific RPA primers (5 μM; 1.2 μL for each primer), reporter probe (5 μM; 1 μL), gene-specific Cas12a gRNAs (25 μM; 0.16 μL for each target region), murine RNase inhibitor (40 U/μL; 0.5 μL), Cas12a (100 μM; 0.16 μL), 10× NEBuffer 2.1 (10 mM Tris-HCl, 50 mM NaCl, 10 mM MgCl₂, 100 μg/mL BSA, pH 7.9 at 1× concentration; 0.5 μL), and reconstituted RPA mix (14.75 μL). After adding NA extract (2 μL) and MgOAc (280 mM, 1.25 μL), we incubated the mix at 42 °C for 20 min, followed by fluorescence measurement (see Supplementary Fig. 19 for the workflow). To analyze mRNA targets, we included 1 U/μL RTase in the CreDiT assay mix. The limit of detection (LOD) was determined based on $S_0 + 3\sigma$, where $S_0$ and $\sigma$ are the signal and the standard deviation of a blank sample, respectively. The limit of quantification (LOQ) was determined based on $S_0 + 10\sigma$. Gel electrophoresis was performed using the Mupid One Electrophoresis System (Eurogentec) for analysis of CreDiT assay products, followed by imaging with the ChemiDoc XRS + System (Bio-Rad). Whole gel images are shown in Supplementary Figs. 20, 21.

## Lyophilization of CreDiT reagents

We prepared the CreDiT assay premix and snap-froze the mixture by immersing it in liquid nitrogen. The frozen mixture was then lyophilized overnight at room temperature on a VirTis FreezeMobile 25 L freeze dryer (SP Scientific). The lyophilized mix was stored at room temperature until use.

## Quantitative PCR (qPCR) and RT-qPCR

We used PCR for quantitative comparison with CreDiT. For DNA targets, qPCR was employed. We prepared the PCR solution (20 μL) by mixing 250 nM IVT primers, 1× Luna Universal qPCR Master Mix, and extracted DNA. We then processed the mixture on a CFX Opus 96 real-time PCR instrument (Bio-Rad). The temperature profile was 95 °C (1 min), followed by cycles of 95 °C (15 s) and 60 °C (30 s). For mRNA targets, we performed RT-qPCR. The sample mixture (20 μL) contains 250 nM gene-specific primers, SuperScript™ III Platinum™ SYBR™ Green One-Step qRT-PCR Kit (Thermo Fisher Scientific), and target RNA (from 8000 cells). The reaction proceeded with the following

protocol: 42 °C, 10 min (RT) and 94 °C, 5 min (RTase inactivation), followed by thermal cycling of 94 °C (5 s), 56 °C (15 s), and 72 °C (15 s). The threshold cycle ($C_t$) was automatically determined by the system software. Gel electrophoresis of the PCR products was run in 2% E-Gel™ EX Agarose Gels on the E-Gel™ Power Snap Electrophoresis Device, followed by imaging with Azure 280 (Azure Biosystems).

## Clinical sample analysis

We collected cervical or anal Pap test samples (BD SurePath™ liquid-based Pap test, Becton, Dickinson and Company) during gynecological or anal exams. Clinical pathologic diagnostics were performed using the BD Onclarity HPV Assay system. Blind to the clinical diagnostic results, we analyzed aliquots (0.5 mL) of the residual clinical materials. We processed samples in the extraction buffer (0.5 mg/mL Pro K in 1× TE) following the heat-based extraction protocol. The extracted NA was then subjected to the CreDiT assay. For each NA target, we used 2 μL of NA extract.

## Statistics and reproducibility

We used GraphPad Prism version 9.5 (GraphPad Software Inc.) or R version 4.2.2 for statistical analyses. For two-group comparisons, we used an unpaired, two-sided $t$-test. For all statistical tests, $p$ values less than 0.05 were considered significant. Details on data presentation and the sample size are included in figure legends. In the study of clinical samples, no statistical method was used to predetermine the sample size, and the experiments were not randomized. No data were excluded from the analyses. The investigators analyzed the clinical samples using the CreDiT assay while blinded to the clinical diagnostic results (gold standard).

## Reporting summary

Further information on research design is available in the Nature Portfolio Reporting Summary linked to this article.

# Data availability

Source data are provided as a Source Data file. Source data are provided with this paper.

# Code availability

The source code for the FWHT signal processing is available at https://doi.org/10.5281/zenodo.12675606[43] under a CC-BY-NC-ND license.

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

## Acknowledgements

This work was supported in part by NIH 1U01CA279858 (C.M.C. and H.L.), U01CA284982 (H.L. and C.M.C.), R01CA229777 (H.L.), R01CA239078 (H.L.), R01HL163513 (H.L.), R01CA237500 (H.L.), R21CA267222 (H.L.), R01CA264363 (C.M.C. and H.L.); 5R25CA174650 (M.P.); MGH Scholar Fund (H.L.); KRIBB Research Initiative Program, KGM5472413 (C.Y.L.); National Research Foundation (Korea) NRF2021R1A6A3A14044562 (H.K.); Ministry of The Ministry of Science and ICT of Korea NRF2022K1A4A8A01080317 (H.G.P.).

## Author contributions

C.Y.L., H.K., I.D., C.M.C., and H.L. designed the study, prepared the figures, and wrote the manuscript. C.Y.L., H.K., I.D., Hanna L., A.S., Y.N., and M.P. conducted the experiments. C.Y.L., H.K., I.D., Hanna L., A.S., Y.N., M.P., H.G.P., and H.L. analyzed the assay data. T.R., A.L., and C.M.C. analyzed clinical data. All authors contributed to the writing of the manuscript.

## Competing interests

The authors declare no competing interests.
