## [Peer Review File · Nature Communications]

REVIEWER COMMENTS

Reviewer #1 (Remarks to the Author):

The authors presented an advanced diagnostic system, also called as CreDiT (CRISPR Enhanced Digital Testing), for rapid on-site detection of nucleic acids (NAs). The CreDiT integrates a one-pot CRISPR strategy to simultaneously amplify target NAs and analytical signals with a robust fluorescent detection method based on digital communication technology. This innovation simplifies NA extraction and detection into a rapid assay (<35 minutes), featuring a straightforward probe design adaptable to new NA targets and a compact device for reliable signal detection. This manuscript is well-structured but could be enhanced by addressing the following points:

- 1、 The manuscript employs enzymatic hydrolysis for nucleic acid releasing, followed by subjecting the system to high temperature (95 degrees) to inactivate the lytic enzyme. In this process, exposure to 95 degrees theoretically suffices for thermal nucleic acid releasing. Why did the author not consider direct thermal lysis for nucleic acid releasing? Has the author compared the efficacy of these two cell lysis methods?
- 2、 The integration of RPA and CRISPR reactions in the same tube raises concerns since activated CRISPR indiscriminately cleaves nucleic acids in its environment. Could this potentially impact the detection results? It is recommended that control experiments be conducted specifically performing RPA amplification followed by CRISPR in the same tube.
- 3、 The description in Table S1 stating a 20-minute analysis time for one-step testing is not very accurate.
- 4、 It is suggested to include synchronous agarose gel images for both Figure 4.b and Figure 4d.
- 5、 For a comparison with standard methods, it is recommended to supplement qPCR data in supplementary materials.

Reviewer #2 (Remarks to the Author):

The authors innovatively developed and assembled a Point-of-Care (POC) prototype named CreDiT for the simultaneous detection of multiple Human Papillomavirus (HPV) strains. The CreDiT platform encompasses two important technological advancements: a one-pot CRISPR strategy that concurrently amplifies target Nucleic Acids (NAs) and assay chemistry; a robust fluorescent detection method founded on digital communication technology. I have the following comments on this work:

1. The authors opted for a pair of Cas12 Ribonucleoproteins (RNP) instead of a single Cas12 RNP to detect RPA amplicons. It would be beneficial if the authors could include a comparison of CreDiT's kinetic performance with varying numbers of Cas12 RNP, such as 1, 2, or even more, to provide a comprehensive understanding of the system's efficiency.
2. The current protocol necessitates CreDiT mix preparation. It would enhance the user-friendliness of the system if the authors could develop premix formulations that can be stored under Point-of-Care conditions. This adjustment would minimize the handling steps involved.
3. In the methods section, the authors need to show the steps to operate the CreDiT assay from an end-user perspective.
4. The authors need to explain in detail how they managed to accomplish one-pot amplification with RPA and detection with Cas12a, especially since Cas12a would be cutting the target in cis and the ssDNA primers in trans, potentially interfering with the reaction. I think their PAMless strategy of detecting ssDNA exposed during amplification overcomes this issue since the target will not be depleted. This design is not new. It was first demonstrated in this paper (PMID: 32948757), but the authors failed to acknowledge it properly.
5. In the discussion section, it would be valuable to incorporate a comparative analysis with previously developed CRISPR-based Point-of-Care devices outlined in prior publications.
6. In the discussion section, it would be valuable to discuss the possibility of automating the handling steps between NA extraction and CreDiT mix preparation.

Reviewer #3 (Remarks to the Author):

Thanks for allowing me to review this interesting paper presenting new HPV testing techniques that have the potential to improve the prevention and management of cervical cancer, a major killer of women, especially in LMICs.

Although I am not a technical expert, I enjoyed reading the paper introducing CreDit, that has the potential of increasing access to molecular testing with a higher sensitivity than conventional PCR.

While development of new, simpler, more accurate and better technologies is key in the prevention of cervical cancer, the main barrier is the feasibility, availability, affordability of screening and treating cervical precancerous lesions in many LMICs, in health systems can cope with the increasing demands and costs for services.

Therefore, I applaud the conclusion of the authors that the new technique has the potential to improve screening and that there are plans to test it in 2 African sites.

Reviewer #3 (Remarks on code availability):

sorry, too complicated

Reviewer #4 (Remarks to the Author):

I thank the writers for their paper. I have a CS/Math background and I am not at all familiar with the field in any sense so please take my review with a grain of salt. I may have misunderstood some things. I was added as a reviewer to review the parts about the Walsh-Hadamard transform.

From what I understand the authors are proposing to use as an excitation waveform a Walsh-Hadamard basis function (a square waveform) as opposed to a vanilla Fourier (sinusoidal) excitation. They claim that this excitation provides a better signal to noise ratio than the normal Fourier excitation (Figure 2b). In general the use of this basis is correct and the explanations are relatively clear.

However, I find it a little bit surprising that this single change would help them outperform a sinusoidal excitation with this large of a margin (Figure 2b). From what I understand this excitation is converted to some sort of LED light. In order to make sure the results are correct I suggest the readers state the "energy" present in the sinusoidal and rectangular waveforms they are using. This can be computed as

the area under the surface of the excitation waveform squared. This would guarantee that in their comparison the LED is emitting the same amount of light in both cases. Currently they are only discussing the amplitude (and not the energy) of the waveform. I could not find a value for the amplitudes in the report either. It is not hard to outperform another excitation, in terms of SNR, if you are using more energy in the excitation signal.

Minor side-comment: You have reported a single sequence number 107 as the optimal sequence index for the excitation signal. However, there is probably a range of sequence indices that are all shifts of one another that are optimal. The optimal index should be something like all integers in the range [99, 120]?

Responses to Reviewers' Comments

Reviewer 1

The authors presented an advanced diagnostic system, also called as CreDiT (CRISPR Enhanced Digital Testing), for rapid on-site detection of nucleic acids (NAs). The CreDiT integrates a one-pot CRISPR strategy to simultaneously amplify target NAs and analytical signals with a robust fluorescent detection method based on digital communication technology. This innovation simplifies NA extraction and detection into a rapid assay (<35 minutes), featuring a straightforward probe design adaptable to new NA targets and a compact device for reliable signal detection. This manuscript is well-structured but could be enhanced by addressing the following points:

1. The manuscript employs enzymatic hydrolysis for nucleic acid releasing, followed by subjecting the system to high temperature (95 degrees) to inactivate the lytic enzyme. In this process, exposure to 95 degrees theoretically suffices for thermal nucleic acid releasing. Why did the author not consider direct thermal lysis for nucleic acid releasing? Has the author compared the efficacy of these two cell lysis methods?

The reviewer raises an excellent point since, indeed, thermal lysis could be used to release nucleic acid (NA) from cells (*Anal Chem* 2023, 95:3476; *Nat Commun* 2022, 13:6480). However, we reasoned *a priori* that using proteinase K (Pro K) could be a superior alternative. Besides lysing cells, Pro K also degrades nuclease proteins (RNases, DNases), potentially improving overall NA-extraction yields. To test this hypothesis, we compared NA extraction between the two methods (Pro K vs. thermal lysis). We processed samples containing similar cell amounts (CaSki, 2.5×10^4 cells/mL). NA was extracted and quantified via qPCR. The results confirmed our hypothesis (**Fig. R1**): the extraction yield of the Pro K method (used in CreDiT) was >6 times higher than that of thermal lysis. Considering that both methods have similar processing times (~16 min) and heating requirements (95 °C), we concluded that using Pro K is a preferred approach for efficient NA release. We have revised the manuscript to clarify this point and included our original comparison data as a new figure (**Supplementary Fig. 8b**).

Fig. R1. Comparison of NA extraction methods. NA was extracted via the CreDiT protocol or thermal lysis (heating at 95 °C). The extracted NA was analyzed for HPV16 by qPCR and HPV16 signal from the CreDiT extraction were used for normalization. The CreDiT protocol led to >6-fold higher NA-extraction yield than thermal lysis. Test samples contained CaSki cells (2.5×10^4 cells/mL). Data are displayed as mean \pm s.d. from technical triplicates.

[Revised Results] Assay optimization for POC operation

When compared to the method using only thermal lysis (95 °C) for NA release, the CreDiT extraction yielded 6 times more NA (**Supplementary Fig. 8b**); this improvement was likely due to the additional benefits of Pro K, notably the degradation of nucleases (i.e., RNases and DNases) that break down NA.

2. The integration of RPA and CRISPR reactions in the same tube raises concerns since activated CRISPR indiscriminately cleaves nucleic acids in its environment. Could this potentially impact the detection results? It is recommended that control experiments be conducted specifically performing RPA amplification followed by CRISPR in the same tube.

The reviewer raises excellent points. We discuss them in two aspects: i) competition between Cas12a and polymerase reactions; and ii) comparison with a two-step approach (RPA followed by Cas12a detection).

i) Cas12a and polymerase competition. As the reviewer pointed out, activated Cas12a becomes an active nuclease, indiscriminately cutting (*trans*-cleavage) *single-stranded* DNA. In particular, these activated Cas12a enzymes can cleave RPA primers (single-stranded oligonucleotides), potentially competing with the DNA replication process. However, kinetic considerations indicate that the polymerase reaction would outpace the cleavage reaction. Below is a brief discussion. For details, please refer to our new **Supplementary Note**.

- The catalytic efficiency (η_t) of Cas12a *trans*-cleavage is defined as $\eta_t = (k_t / K_{Mt})$, where k_t is the catalytic turnover rate and K_{Mt} is the Michaelis-Menten (MM) constant. We experimentally measured $k_t = 1.2 \text{ s}^{-1}$ and $K_{Mt} = 3.4 \text{ }\mu\text{M}$ for our Cas12a system, which yielded $\eta_t = 3.5 \times 10^5 \text{ s}^{-1} \text{ M}^{-1}$. The reaction rate (a_t) is then given as $a_t = \eta_t \cdot C_a$, where C_a is the concentration of activated Cas12a enzyme. We conservatively assume that all Cas12a enzyme in a sample is activated ($C_a = C_0 = 0.64 \text{ }\mu\text{M}$), which gives $a_t = 0.2 \text{ s}^{-1}$.
- The RPA kit uses polymerase I, whose kinetic parameters are estimated to be $k_p = 50 \text{ s}^{-1}$ and $K_{Mp} = 1.3 \text{ }\mu\text{M}$ (*Int. J. Mol. Sci.* **23**, 6373). k_p is the catalytic turnover rate and K_{Mp} is the MM constant for the polymerase. From these values, the catalytic efficiency (η_p) of polymerase is calculated as $\eta_p = (k_p / K_{Mp}) = 3.8 \times 10^7 \text{ s}^{-1} \text{ M}^{-1}$. The polymerase concentration in the RPA was $1.3 \text{ }\mu\text{M}$ (*Int. J. Mol. Sci.* **23**, 6373). Therefore, the reaction rate (a_p) for polymerase is $a_p = 52 \text{ s}^{-1}$.

	Enzyme concentration (μM)	Turnover rate (s^{-1})	MM constant (μM)	Reaction rate (s^{-1})
CAS12a trans	$C_0 = 0.64$	$k_t = 1.2$	$K_{Mt} = 3.4$	$a_t = 0.2$
Polymerase	$P_0 = 1.3$	$k_p = 50$	$K_{Mp} = 1.3$	$a_p = 52$

Given this significant (>200-fold) difference in reaction rates, the polymerase is expected to be more dominant than the Cas12a *trans*-cleavage activity, ensuring sufficient DNA amplification during the CreDiT assay to generate actionable readouts.

ii) Comparison with two-step assay. As suggested by the reviewer, we performed a two-step assay, wherein the RPA (20 min) and Cas12a (30 min) reactions were carried out sequentially. We followed the previously reported DETECTR protocol (*Science* **360**, 436). In this two-step assay, the Cas12a was needed to detect double-stranded DNA. Therefore, we designed a new gRNA specific to the HPV16 target DNA sequence adjacent to the PAM site (TTTN). The titration experiments (**Fig. R2**) demonstrated a superior performance of the CreDiT assay – it achieved a lower detection limit (40 copies/mL) than the two-step assay (detection limit, 400 copies/mL) while affording a shorter assay time (20 min vs. 50 min, respectively). These results reaffirm CreDiT's practical advantage for rapid, sensitive NA detection.

Fig. R2. Comparison between CreDiT and a two-step assay. Serially diluted HPV16 target DNA samples were analyzed by CreDiT (20 min) and a two-step assay in which RPA (20 min) was followed by Cas12a reaction (30 min). CreDiT exhibited a lower detection limit (40 copies/mL) than the two-step assay (detection limit, 400 copies/mL). ΔCreDiT and $\Delta\text{Two-step}$ are the background-subtracted signals. Data are displayed as mean \pm s.d. from technical triplicates.

iii) Revisions made. We have incorporated the following changes.

- The new **Supplementary Note** presents details about i) experimentally measuring the catalytic efficiency of Cas12a *trans*-cleavage and ii) the kinetics consideration of Cas12a and polymerase competition. For the reviewer's convenience, we have attached the Note at the end of this response document.

- The new data (shown in **Fig. R2**) is presented in **Supplementary Fig. 13**.
- The main text was revised to include new discussions on kinetic measurements.

[Revised Results] CreDiT platform development

During CreDiT's initial phase, the polymerase enzyme separates the double-stranded target DNA, creating a single-stranded region for the gRNA molecule to recognize. This recognition triggers the Cas12a enzyme within the Cas12a/gRNA complex to act as a nuclease and cleave the F-Q DNA probes, generating a fluorescent signal. However, activated Cas12a can also interfere with RPA by cleaving the target DNA itself (*cis*-cleavage) and breaking down the RPA primers (*trans*-cleavage). Reassuringly, our kinetic analysis indicates that RPA is faster than these competing reactions (see **Supplementary Note** for details). This dominance of RPA ensures sufficient amplification of the target DNA within the CreDiT format.

[Revised Results] CreDiT probe design and validation

Next, we compared CreDiT's performance with the established gold standard, qPCR, and a two-step assay, where RPA (20 min) and Cas12a detection (30 min) were carried out sequentially. We used synthetic HPV16 DNA as a model target. CreDiT demonstrated superior sensitivity compared to qPCR (**Fig. 4c**) and the two-step assay (**Supplementary Fig. 13**). CreDiT's limit of detection (LOD) was down to 40 copies/mL, whereas LODs for qPCR and the two-step assay were around 400 copies/mL. CreDiT also offered an advantage in assay time, requiring only 20 minutes compared to 1 hour for qPCR and 50 minutes for the two-step assay.

3. The description in Table S1 stating a 20-minute analysis time for one-step testing is not very accurate.

We apologize for this oversight. The term "assay time" in **Supplementary Table 1** was intended to denote the duration for nucleic acid (NA) detection, excluding the time for NA extraction. We made this decision as the information on NA extraction was often missing in the cited references. To clarify this point, we have revised the column title to "NA detection time (min)". Additionally, we have explicitly stated that this metric only pertains to the time required for NA detection.

Supplementary Table 1. Comparison with CRISPR-based HPV molecular tests.

System	Readout	NA detection time (min) [†]	LOD	Assay characteristics	Reference
Electrochemically active electrode	Electrochemistry	70	50 pM	 • Specialized chip fabrication • Low sensitivity • Narrow target coverage • Absence of clinical study 	1
Electrochemically active electrode	Electrochemistry	60	30 pM	 • Specialized chip fabrication • Low sensitivity • Narrow target coverage 	2
...					
RPA+Cas					
CreDiT (Our work)	Fluorescence	20	66 zM	 • Single step • Broad target coverage 	-

[†] This metric only considers the duration for nucleic acid (NA) detection, excluding the time for NA extraction.

4. It is suggested to include synchronous agarose gel images for both Figure 4b and Figure 4d.

As the reviewer suggested, we conducted gel electrophoresis experiments to support the results shown in **Figs. 4b and 4d**. The gel images are now shown in **Supplementary Figs. 11 and 15**, and the protocol is reported in the Methods section.

Supplementary Figure 11. Gel electrophoresis analysis on the CreDiT assay products. (a) HPV16 DNA was used as a model target with a concentration of 4×10^7 copies/mL. Lanes 1 and 2 were RPA controls, confirming target-specific DNA amplification (red rectangle). Lane 3 was the control for Cas12a/gRNA with excess amounts of Cas12a (5.12 μ M) and gRNA (1.28 μ M) applied. Lanes 4 and 5 had the RPA reaction conditions in the presence of Cas12a (640 nM) or gRNA (160 nM), respectively. Lanes 6 and 7 had the complete CreDiT reaction condition without (Lane 6) or with (Lane 7) the target DNA input. The RPA reactions successfully produced DNA amplicons (white rectangles) regardless of the presence of Cas12a, gRNA, or both. **(b)** For the assay conditions in Lanes 4 to 7 in (a), we further confirmed the cleavage of reporter probes, i.e., single-stranded DNAs tagged with a fluorescent dye (F) and quencher (Q) pair. Only Lane 7, which had the complete CreDiT reaction in the presence of the target DNA, showed the fluorescent signal (green rectangle) from the cleaved reporters. Lane 8 was from a control sample containing F-Q DNA probes only. The gel was stained for nucleic acids with GelRed® (Biotium). Two separate fluorescent images were acquired from the same gel, detecting GelRed and the reporter fluorophore (fluorescein amidite). The two images were then merged; whole gel images are shown in **Supplementary Fig. 19**.

[†] Cas12a (640 nM), gRNAs (60 nM), RPA primers (240 nM)

Supplementary Figure 15. Probe specificity. CreDiT assays were performed using probes designed to detect various hrHPV targets. The corresponding electrophoretic results show that target DNA (red rectangles) was amplified only when its matching probes were present in the CreDiT. The results confirmed the high specificity of the CreDiT assay. Lane 3 is the positive control wherein the target DNA was amplified via RPA only; amplicons are indicated by black rectangles. The initial concentration of an input DNA was 4×10^7 copies/mL for all reactions. The whole gel images are shown in **Supplementary Fig. 20**.

5. For a comparison with standard methods, it is recommended to supplement qPCR data in supplementary materials.

We fully concur with the reviewer. During our assay development, we indeed compared CreDiT results with qPCR (Fig. R3). Please refer to Fig. 4c (for DNA targets) and Supplementary Fig. 16b (for mRNA targets). For the clinical samples, qPCR was performed at a Massachusetts General Hospital clinical pathology laboratory using the BD Onclarity HPV Assay system. This assay reports binary results (HPV-target positive vs. negative) based on pre-determined cutoffs. The new Supplementary Table 4 shows these Onclarity results along with the CreDiT data.

Fig. R3. Comparison between CreDiT and qPCR. (Left) DNA detection shown in Fig. 4c. (Right) mRNA detection reported in Supplementary Fig. 16b.

Supplementary Table 4. Clinical qPCR and CreDiT results for cervical cancer screening.

Sample ID	Anatomic source	Onclarity results (qPCR) [†]	CreDiT results (a.u.)						
			GAPDH	HPV16	HPV18	HPV45	HPV31	HPV33	HPV58
1	Cervical	HPV16	3.41	8.34	0.02	0.00	0.04	0.00	0.00
2	Cervical	HPV16	3.08	0.02	0.00	0.00	0.04	0.01	0.02
3	Cervical	HPV16	10.65	2.55	0.00	0.00	0.00	0.00	0.00
4	Vaginal	HPV16	12.72	2.17	0.04	0.00	0.00	0.03	0.00
...									
55	Cervical	HPV45	13.96	0.00	0.03	0.19	0.00	0.00	0.02
56	Cervical	HPV45	7.53	0.02	0.00	0.07	0.00	0.00	0.00
57	Cervical	HPV16, 18, 45, other [‡]	3.46	0.12	4.11	0.63	0.00	0.00	0.00
58	Cervical	HPV16, 18, other	7.75	21.89	0.92	0.00	0.00	0.00	0.00
59	Cervical	HPV16, other	1.60	0.19	0.00	0.01	0.09	0.00	0.00
...									
117	Cervical	Negative	6.13	0.00	0.03	0.00	0.08	0.00	0.01
118	Cervical	Negative	7.30	0.00	0.00	0.00	0.00	0.00	0.00
119	Cervical	Negative	17.36	0.01	0.00	0.05	0.00	0.00	0.00
120	Cervical	Negative	13.18	0.00	0.00	0.00	0.00	0.01	0.00
121	Cervical	other	1.95	0.03	0.00	0.00	0.05	18.73	0.00

[†] The reported limit of detection (LOD) is 251 (HPV16), 1083 (HPV18), 1216 (HPV45), 830 (HPV31), 1665 (HPV33), and 2369 (HPV58) copies/mL. The cutoff C_t values are 38.3 for HPV16 and 34.2 for other targets.

[‡] "Other" includes HPV31, 33, 35, 39, 51, 52, 56, 58, 59, 66, and 68.

Reviewer 2

The authors innovatively developed and assembled a Point-of-Care (POC) prototype named CreDiT for the simultaneous detection of multiple Human Papillomavirus (HPV) strains. The CreDiT platform encompasses two important technological advancements: a one-pot CRISPR strategy that concurrently amplifies target Nucleic Acids (NAs) and assay chemistry; a robust fluorescent detection method founded on digital communication technology. I have the following comments on this work:

1. The authors opted for a pair of Cas12 Ribonucleoproteins (RNP) instead of a single Cas12 RNP to detect RPA amplicons. It would be beneficial if the authors could include a comparison of CreDiT's kinetic performance with varying numbers of Cas12 RNP, such as 1, 2, or even more, to provide a comprehensive understanding of the system's efficiency.

We thank the reviewer for this excellent suggestion. As recommended, we have compared the assay kinetics while varying the number of Cas12a binding sites. We designed three gRNAs targeting distinct regions of *HPV16* DNA (**Fig. R4a**): forward (F), reverse (R), and middle (M). We then performed the CreDiT assay using these gRNAs in various combinations (**Fig. R4b**) while keeping the total gRNA concentration constant ($0.48 \mu\text{M}$). The assay mix included RPA reagents with forward and reverse primers. We made the following observations.

- **Individual gRNA performance.** Among individual gRNAs, F-gRNA produced the strongest signal, followed by R-gRNA. M-gRNA displayed the weakest response. These variations likely arise from differences in the efficiency of gRNA binding to their target sequences.
- **Synergy of F & R gRNAs.** The combination of F-gRNA and R-gRNA (the CreDiT protocol) generated the strongest analytical signal. Notably, this combined signal was close to the sum of the signals using individual gRNAs. This observation strongly supports the benefit of targeting both the F and R regions, as employed in the CreDiT protocol.
- **Impact of M-gRNA.** Interestingly, using all three gRNAs (F, R, and M) resulted in lower signals compared to the dual-probe configuration (F and R-gRNA). With the total gRNA amount fixed, adding the less efficient M-gRNA likely reduces the overall Cas12a activity.

We have revised the main text to incorporate this discussion. The new data shown in **Fig. R4** is presented in **Supplementary Fig. 12**.

Fig. R4. Impact of gRNA combinations on CreDiT assay signals. (a) Three gRNAs were designed to target distinct regions of the HPV16 DNA: forward (F), reverse (R), and middle (M). (b) The CreDiT assay was performed using various combinations of these gRNAs. The combination of F-gRNA and R-gRNA (the CreDiT protocol) yielded the strongest analytical signal. Notably, this combined signal was close to the sum of the signals obtained using individual gRNAs. Incorporating the M-gRNA, which displayed the weakest signal as an individual gRNA, resulted in a lower overall CreDiT signal. Because the total gRNA amount was fixed (480 nM), adding the less efficient M-gRNA likely reduces the availability of F-gRNA and R-gRNA for Cas12a binding. [HPV16 DNA] = 4×10^7 copies/mL. ΔCreDiT is the background-subtracted signal. Data are displayed as mean from technical triplicates.

[Revised Results] CreDiT probe design and validation

We validated the designed probes in the CreDiT assay. We first varied the composition of the assay reagents and monitored the resulting CreDiT signals (**Fig. 4b** and **Supplementary Fig. 11**). As expected, the fluorescent signal was only observed when all CreDiT reagents and the target DNA were present. We further observed the synergistic benefit of recognizing both the forward and reverse regions in a target DNA. The combination of forward and reverse gRNAs produced stronger analytical signals than individual gRNAs (**Supplementary Fig. 12**). We also designed an additional gRNA that recognized a middle region of the target DNA (**Supplementary Table 2**). However, this gRNA displayed the weakest signal generation, and incorporating it into the assay lowered the overall signal, likely due to competition with the more efficient forward and reverse gRNAs for Cas12a binding.

2. The current protocol necessitates CreDiT mix preparation. It would enhance the user-friendliness of the system if the authors could develop premix formulations that can be stored under Point-of-Care conditions. This adjustment would minimize the handling steps involved.

We fully agree with the reviewer that a premix format will enhance user-friendliness. As such, we prepared a lyophilized premix and tested its applicability (**Fig. 3e**). We found that this powdered premix retained the assay efficiency of freshly prepared reagents for at least two weeks. We have revised the main text to highlight this point and incorporated the lyophilization protocol in the Methods section.

[Revised Results] Assay optimization for POC operation

Regarding CreDiT reagents, we premixed the components (RPA primers, reporter probe, Cas12a gRNAs, Cas12a protein) and lyophilized the mixture (see **Methods**) to facilitate its transport and storage. When evaluated via the CreDiT assay, the lyophilized premix retained its efficacy for at least two weeks in storage at ambient conditions (**Fig. 3e**). Overall, these features could mitigate logistic challenges and promote CreDiT usability in resource-limited settings even without cold chain solutions.

[Revised Methods] Lyophilization of CreDiT reagents

We prepared the CreDiT assay premix as described above and snap-froze the mixture by immersing it in liquid nitrogen. The frozen mixture was then lyophilized overnight at room temperature on a VirTis FreezeMobile 25L freeze dryer (SP Scientific). The lyophilized mix was stored at room temperature until use.

Fig. 3e. CreDiT reagents were lyophilized and stored under ambient conditions. The lyophilized reagents demonstrated consistent activity for at least two weeks.

3. In the methods section, the authors need to show the steps to operate the CreDiT assay from an end-user perspective.

This is an insightful and impactful point. We have carefully revised the Methods section to present detailed assay steps. Furthermore, we have prepared an assay flowchart (new **Supplementary Fig. 18**) as a quick reference guide for end users.

Supplementary Fig. 18. CreDiT workflow.

4. The authors need to explain in detail how they managed to accomplish one-pot amplification with RPA and detection with Cas12a, especially since Cas12a would be cutting the target in *cis* and the ssDNA primers in *trans*, potentially interfering with the reaction. I think their PAMless strategy of detecting ssDNA exposed during amplification overcomes this issue since the target will not be depleted. This design is not new. It was first demonstrated in this paper (PMID: 32948757), but the authors failed to acknowledge it properly.

The reviewer correctly highlighted the concern that Cas12a activity could hinder RPA. Activated Cas12a enzymes can interfere with RPA by degrading both DNA targets (*cis*-cleavage) and RPA primers (*trans*-cleavage). However, our CreDiT assay successfully generates sufficient DNA amplicons within a single reaction. To understand this observation, we compared the reaction rates of three key processes: DNA replication by RPA, Cas12a-mediated *cis*-cleavage, and *trans*-cleavage. In essence, the RPA reaction is dominant over the interfering Cas12a activities, enabling the one-pot CreDiT assay. A detailed explanation is provided in our new **Supplementary Note**. Below is its brief summary.

i) RPA vs. Cas12a's *cis*-cleavage. The activated Cas12a can cleave its targeted DNA, which makes the DNA strand incompatible with RPA. We first estimated the catalytic efficiency values of RPA and Cas12a *cis*-cleavage. The catalytic efficiency is defined as $\eta = (k / K_M)$, where k is the catalytic turnover rate and K_M is the Michaelis-Menton (MM) constant. The reaction rate (a) is then calculated by multiplying η with the enzyme concentration. Using k and K_M values from the literature, we have the following estimation for our CreDiT assay:

	Enzyme concentration (μM)	Turnover rate (s ⁻¹)	MM constant (μM)	Reaction rate (s ⁻¹)
CAS12a cis	$C_0 = 0.64$	$k_c = 0.005$	$K_{Mc} = 5.2 \times 10^{-4}$	$a_c = 6.2$
Polymerase	$P_0 = 1.3$	$k_p = 50$	$K_{Mp} = 1.26$	$a_p = 52$

DNA production by polymerase (a_p) is about 9 times faster than DNA degradation by Cas12a *cis*-cleavage (a_c), which indicates that the DNA amount will exponentially increase over time.

ii) RPA vs. Cas12a's *trans*-cleavage. Activated Cas12a can hinder DNA replication by indiscriminately cutting (*trans*-cleavage) RPA primers (single-stranded oligonucleotides). We followed a similar kinetic analysis as above to compare the reaction rates between Cas12a *trans*-cleavage and DNA replication.

	Enzyme concentration (μM)	Turnover rate (s^{-1})	MM constant (μM)	Reaction rate (s^{-1})
CAS12a trans	$C_0 = 0.64$	$k_t = 1.2$	$K_{Mt} = 3.4$	$a_t = 0.2$
Polymerase	$P_0 = 1.3$	$k_p = 50$	$K_{Mp} = 1.3$	$a_p = 52$

Given this significant (>200-fold) difference in reaction rates, the polymerase is expected to outpace the Cas12a *trans*-cleavage activity, ensuring sufficient DNA amplification during the CreDiT assay.

iii) Revisions. We have made the following changes to address the reviewer's comment.

- A new **Supplementary Note** is incorporated, presenting i) experiments that measured the catalytic efficiency of Cas12a *trans*-cleavage and ii) the kinetics considerations of Cas12a and polymerase competition. We attached the Note at the end of this response document.
- The referenced paper (PMID: 32948757) is cited and acknowledged.
- The main text was revised to include new discussions on kinetic measurements.

[Revised Results] CreDiT platform development

During CreDiT's initial phase, the polymerase enzyme separates the double-stranded target DNA, creating a single-stranded region for the gRNA molecule to recognize. This recognition triggers the Cas12a enzyme within the Cas12a/gRNA complex to act as a nuclease and cleave the F-Q DNA probes, generating a fluorescent signal. However, activated Cas12a can also interfere with RPA by cleaving the target DNA itself (*cis*-cleavage) and breaking down the RPA primers (*trans*-cleavage). Fortunately, our kinetic analysis indicates that RPA is faster than these competing reactions (see **Supplementary Note** for details). This dominance of RPA ensures sufficient amplification of the target DNA within the CreDiT format.

All assay steps are performed at a fixed temperature (42 °C, isothermal) without requiring external interruptions. Furthermore, the assay is fast and sensitive, as it concurrently amplifies target DNA (RPA) and analytical signals (Cas12a/gRNA). Similar one-pot approaches have been described^{16,17}; our analysis may be applicable to understand their underlying reaction mechanisms and inform optimizations.

5. In the discussion section, it would be valuable to incorporate a comparative analysis with previously developed CRISPR-based Point-of-Care devices outlined in prior publications.

This is an excellent suggestion offering contextual impact; we revised our discussion, accordingly. We have also expanded our **Supplementary Table 1**, listing other CRISPR-based point-of-care systems and comparing their performances with CreDiT.

[Revised Discussions]

These technical strengths (sensitivity, throughput, integration, and ease of use) may differentiate CreDiT from other Cas-based point-of-care systems that require microfluidic components²⁸⁻³⁰, rely on microscopy³¹, or produce semi-quantitative results³² (see **Supplementary Table S1** for comparison).

Supplementary Table 1. Comparison with CRISPR-based HPV molecular tests.

System	Readout	NA detection time (min) [†]	LOD	Assay characteristics	Reference
Electrochemically active electrode	Electrochemistry	70	50 pM	 Specialized chip fabrication Low sensitivity Narrow target coverage Absence of clinical study 	1
Electrochemically active electrode	Electrochemistry	60	30 pM	 Specialized chip fabrication Low sensitivity Narrow target coverage Absence of clinical study 	2
Electrochemiluminescence-active electrode	Electrochemiluminescence	70	0.48 pM	 Specialized chip fabrication Low sensitivity Narrow target coverage Absence of clinical study 	3
Lateral flow assay; RPA+Cas	Color	>180	0.24 fM	 Need for several steps and reagent Subjective, qualitative results Narrow target coverage Long assay time 	4
Lateral flow assay; LAMP+Cas	Color	60	N/A	 Need for several steps and reagent Subjective, qualitative results Complicated probe design Narrow target coverage 	5
Lateral flow assay; RPA+Cas	Color	105	3.3 aM	 Specialized chip fabrication Need for several steps and reagents Need for a specialized reporter Complicated probe design Narrow target coverage 	6
Lateral flow assay; RPA+Cas	Color	30	1 aM	 Specialized chip fabrication Need for several steps and reagents Subjective, qualitative results Narrow target coverage 	7
Dynamic aqueous multiphase reaction; RPA+Cas	Fluorescence	60	5 aM	 Narrow target coverage 	8
CDetection; RPA+Cas	Fluorescence	190	1 aM	 Involving two steps Narrow target coverage Long assay time 	9
DETECTR; RPA+Cas	Fluorescence	70	Attomolar	 Involving two steps Narrow target coverage 	10
Polydisperse droplet digital assay	Fluorescence	30	100 aM	 Narrow target coverage 	11
Microfluidic dual-droplet device; RPA+Cas	Fluorescence	30	1 aM	 Specialized chip fabrication and signal measurement (i.e. imaging) Delicate care of droplets Narrow target coverage 	12
MiCaR; RPA+Cas	Fluorescence	>40	1.7 fM	 Specialized chip fabrication and signal measurement (i.e. imaging) Complicated assay preparation 2 steps 	13
DROPT; RPA+Cas	Fluorescence	30	1 aM	 Narrow target coverage 2 steps 	14
CreDiT (Our work)	Fluorescence	20	66 zM	 Single step Broad target coverage 	-

[†] This metric only considers the duration for nucleic acid (NA) detection, excluding the time for NA extraction.

6. In the discussion section, it would be valuable to discuss the possibility of automating the handling steps between NA extraction and CreDiT mix preparation.

This is a good constructive point. Currently, the CreDiT assay requires the manual step of adding the CreDiT mix after NA extraction. To streamline this process and enhance user-friendliness, we propose a new sample tube design (**Fig. R5**). This tube would feature two compartments separated by a readily-puncturable seal. The top chamber will be pre-filled with CreDiT reagents, while the bottom compartment will house a pre-loaded aliquot of the NA extraction buffer. This dual-chamber design will safeguard the CreDiT reagents throughout the NA extraction process. Following extraction, the CreDiT mix can be effortlessly introduced into the reaction chamber by simply breaking the seal – an action readily amenable to automation via a basic plunger push. We have incorporated this discussion in the revised manuscript.

Fig. R5. New tube design for automation. The tube features two compartments separated by a breakable seal. The top compartment is pre-loaded with an "all-in-one" CreDiT reagent mix, while the bottom compartment is pre-filled with the NA extraction buffer. Following NA extraction, the seal is broken to introduce the CreDiT reagents into the sample located in the bottom chamber.

[Revised Discussions]

We could further streamline the assay flow. In its current format, CreDiT requires a manual step to add the CreDiT mix into a sample after NA extraction. This step can be simplified with a new sample tube design. This tube would feature two compartments separated by a breakable seal. The top compartment will be pre-filled with CreDiT reagents, while the bottom will house the NA extraction buffer. This dual-chamber design will protect the CreDiT reagents throughout the NA extraction process. Following NA extraction, the CreDiT mix can be effortlessly introduced into the reaction chamber by breaking the seal – an action readily amenable to automation or manual operation.

Reviewer 3

Thanks for allowing me to review this interesting paper presenting new HPV testing techniques that have the potential to improve the prevention and management of cervical cancer, a major killer of women, especially in LMICs.

Although I am not a technical expert, I enjoyed reading the paper introducing CreDiT, that has the potential of increasing access to molecular testing with a higher sensitivity than conventional PCR.

While development of new, simpler, more accurate and better technologies is key in the prevention of cervical cancer, the main barrier is the feasibility, availability, affordability of screening and treating cervical precancerous lesions in many LMICs, in health systems can cope with the increasing demands and costs for services.

Therefore, I applaud the conclusion of the authors that the new technique has the potential to improve screening and that there are plans to test it in 2 African sites.

We highly appreciate the positive evaluation and supportive comments on this work.

Reviewer #4 (Remarks to the Author):

I thank the writers for their paper. I have a CS/Math background and I am not at all familiar with the field in any sense so please take my review with a grain of salt. I may have misunderstood some things. I was added as a reviewer to review the parts about the Walsh-Hadamard transform. From what I understand the authors are proposing to use as an excitation waveform a Walsh-Hadamard basis function (a square waveform) as opposed to a vanilla Fourier (sinusoidal) excitation. They claim that this excitation provides a better signal to noise ratio than the normal Fourier excitation (Figure 2b). In general the use of this basis is correct and the explanations are relatively clear.

1. However, I find it a little bit surprising that this single change would help them outperform a sinusoidal excitation with this large of a margin (Figure 2b). From what I understand this excitation is converted to some sort of LED light. In order to make sure the results are correct I suggest the readers state the "energy" present in the sinusoidal and rectangular waveforms they are using. This can be computed as the area under the surface of the excitation waveform squared. This would guarantee that in their comparison the LED is emitting the same amount of light in both cases. Currently they are only discussing the amplitude (and not the energy) of the waveform. I could not find a value for the amplitudes in the report either. It is not hard to outperform another excitation, in terms of SNR, if you are using more energy in the excitation signal.

The reviewer raised an excellent point regarding the impact of input energy on signal-to-noise ratio (SNR). In response, we subsequently conducted additional analyses considering the energy present in the sinusoidal and Walsh waveforms used for LED excitation. To power the LED within the full voltage swing (0 – 6 V), we adjusted the excitation waveform by scaling it and adding a positive voltage offset. For the Walsh modulation function (W), the input voltage was $V_w = 3 \cdot W + 3$ (V). Similarly, for sinusoidal excitation, the input voltage was $V_s = 3 \cdot \sin(2\pi/T \cdot t) + 3$ (V), where T is the waveform period. The average power to the LED is proportional to the square of the input voltage. For the Walsh modulation, the power p_w is

$$\begin{aligned} p_w &\propto \frac{1}{T} \int_0^T |V_w|^2 dt = \frac{9}{T} \int_0^T (W + 1)^2 dt \\ &= \frac{9}{T} \int_0^T (W^2 + 2W + 1) dt = \frac{9}{T} \cdot 2T = 18 \end{aligned}$$

For the sinusoidal modulation, the power p_s is

$$\begin{aligned} p_s &\propto \frac{1}{T} \int_0^T |V_s|^2 dt = \frac{9}{T} \int_0^T [\sin(2\pi/T \cdot t) + 1]^2 dt \\ &= \frac{9}{T} \int_0^T [\sin^2(2\pi/T \cdot t) + 1] dt = \frac{9}{T} \cdot \frac{3}{2} T = 13.5 \end{aligned}$$

The observed SNR values were 166 for Walsh and 98 for Fourier modulations (**Fig. 2b**). Using values of Fourier modulation as a scaler, we estimated ratios of SNR, input power, and power-normalized SNR (**Table R1**).

This analysis revealed that, for a given input power, Walsh modulation still yielded a higher SNR than Fourier modulation, which further confirmed the advantage of using the Walsh method in our signal processing. We have revised the Method section to clarify the device operation and present this new analysis.

Modulation	SNR	Input power (p)	SNR/ p
Walsh	1.7	1.3	1.3
Fourier	1	1	1

Table R1. SNR and power comparison between Walsh and Fourier modulations.

[Revised Methods] CreDiT signal processing for fluorescent measurements

For fluorescent measurements, we used Walsh functions of size 512. To generate the excitation waveform $E(t)$, we specified an input vector D of length 512. The i -th element of D represented the value for the i -th sequency in the WH domain. Applying the Fast Walsh-Hadamard transform (FWHT) to D yielded the waveform $E(t)$. We adjusted the excitation waveform by scaling it and adding a positive voltage offset to drive an LED in the full voltage swing (0 – 6 V). The resulting input voltage was $E(t) = 3 \cdot W(t) + 3$ (V), where $W(t)$ is the Walsh function. We subsequently measured a Walsh-modulated fluorescence signal using a photodiode. This signal $S(t)$ was directly digitized (200 kHz) by a 12-bit analog-to-digital converter built into the MCU (MK20DX256). The MCU then performed the inverse FWHT, converting $S(t)$ into the sequency vector T . We finally obtained the signal intensity by taking an inner vector product of $D \cdot T$. In the current CreDiT prototype, we used an input vector D_{107} , whose 107th element was the only non-zero element. This sequency was close to the operating frequency of 20.9 kHz. A similar approach was applied for the Fourier modulation, with the input voltage $E(t) = 3 \cdot \sin(2\pi f \cdot t) + 3$ (V), where f is the waveform frequency. The average input power to LED had a 4:3 ratio between Walsh and Fourier modulations.

2. Minor side-comment: You have reported a single sequency number 107 as the optimal sequency index for the excitation signal. However, there is probably a range of sequency indeces that are all shifts of one another that are optimal. The optimal index should be something like all integers in the range [99, 120]?

This is indeed an astute point. To determine the optimal sequency, we evaluated our system's performance under the excitation of different Walsh waveforms. Specifically, we measured the fluorescent signals from a sample while sweeping Walsh sequency (New **Supplementary Fig. 5a**). Our results indicated that the system achieved high SNR with sequences between 97 and 130 (**Supplementary Fig. 5b**); we selected sequency 107 for its highest SNR in our current prototype. However, we note that users can readily adjust the sequency number based on their system's hardware to maximize SNR. These points and new data are now incorporated in the revised manuscript.

[New] Supplementary Fig. 5. Determining the optimal Walsh sequency for excitation signals. (a)

Fluorescent intensities were measured using the CreDiT system, while the Walsh sequency was swept from 0 to 512. The mean fluorescence (green) is the average from 60 wavetrains at each sequency. The standard deviation is shown as a 95% confidence interval (blue). (b) The signal-to-noise ratio (SNR) was computed based on the measured data. The system achieved high SNR within the Walsh sequency range of 97 and 130 (gray shade). The peak SNR was observed at the sequency 107.

SUPPLEMENTARY NOTE – REACTION KINETICS ESTIMATION

List of symbols

	Description	Unit
D	concentration of DNA target	M
d	concentration of cleaved DNA target	M
Cas12a trans-cleavage		
C	concentration of Cas12a	M
C_a	concentration of Cas12a and substrate complex	M
C_e	concentration of Cas12a and substrate complex after cis -cleavage	M
C_0	initial concentration of Cas12a	M
k_{1c}	forward reaction constant for Cas12a cis -cleavage	$s^{-1} \cdot M^{-1}$
k_{2c}	backward reaction constant for Cas12a cis -cleavage	s^{-1}
k_c	catalytic turnover rate for Cas12a cis -cleavage	s^{-1}
K_{Mc}	Michaelis-Menten constant for Cas12a cis -cleavage	M
k_{1t}	forward reaction constant for Cas12a trans -cleavage	$s^{-1} \cdot M^{-1}$
k_{2t}	backward reaction constant for Cas12a trans -cleavage	s^{-1}
k_t	catalytic turnover rate for Cas12a trans -cleavage	s^{-1}
K_{Mt}	Michaelis-Menten constant for Cas12a trans -cleavage	M
Polymerase reaction		
P	concentration of polymerase	M
P_a	concentration of polymerase and substrate complex	M
P_0	initial concentration of polymerase	M
k_{1p}	forward reaction constant for polymerase	$s^{-1} \cdot M^{-1}$
k_{2p}	backward reaction constant for polymerase	s^{-1}
k_p	catalytic turnover rate for polymerase	s^{-1}
K_{Mp}	Michaelis-Menten constant for polymerase	M

The CreDiT assay relies on two concurrent reactions: i) target DNA amplification by RPA and ii) cleavage of signal probes by Cas12a. However, Cas12a activity can impede RPA because activated Cas12a enzyme can degrade both its DNA target (*cis*-cleavage) and RPA primers (*trans*-cleavage). Interestingly, our CreDiT results suggest that RPA outpaces the cleavage reactions, enabling efficient DNA replication within a single reaction mixture. To support this observation, we estimated and compared the reaction rates of these three processes: DNA replication, *cis*-, and *trans*-cleavages.

1. DNA replication and *cis*-cleavage

We consider two simplified reactions: i) a polymerase reaction that amplifies DNA and iii) a single turnaround DNA cleavage by Cas12a. The governing reactions are written as

These two reactions compete with each other: Cas12a destroys DNA (D), while the polymerase creates a new DNA molecule. The related kinetic equations are

$$(1) \quad \frac{dC_a}{dt} = k_{1c}CD - k_{2c}C_a - k_cC_a$$

$$(2) \quad \frac{dC}{dt} = -k_{1c}CD + k_{2c}C_a$$

$$(3) \quad \frac{dP_a}{dt} = k_{1p}PD - k_{2p}P_a - k_pP_a$$

$$(4) \quad \frac{dP}{dt} = -k_{1p}PD + k_{2p}P_a + k_pP_a$$

$$(5) \quad \frac{dD}{dt} = -k_{1c}CD + k_{2c}C_a - k_{1p}PD + k_{2p}P_a + 2k_pP_a$$

We consider the quasi-equilibrium state wherein the formation and breakdown of the enzyme-substrate complex are in dynamic equilibrium; this state is reached shortly after the initial transient period when the reaction starts. For these steady states, we can approximate

$$(6) \quad \begin{cases} \frac{dC_a}{dt} = k_{1c}CD - k_{2c}C_a - k_cC_a = 0 \\ \frac{dP_a}{dt} = k_{1p}PD - k_{2p}P_a - k_pP_a = 0 \end{cases}$$

which leads to

$$(7) \quad \begin{cases} C_a = \frac{k_{1c}}{k_{2c} + k_c}CD = \frac{CD}{K_{Mc}} \\ P_a = \frac{k_{1p}}{k_{2p} + k_p}PD = \frac{PD}{K_{Mp}} \end{cases}$$

where K_{Mc} and K_{Mp} are the Michaelis-Menten (MM) constant for Cas12a *cis*-cleavage and the polymerase reaction, respectively. Using C_a and P_a in Eq. (7), we simplify Eq (5) as

$$(8) \quad \frac{dD}{dt} = -\frac{k_c}{k_{Mc}}CD + \frac{k_p}{k_{Mp}}PD.$$

The first term on the right side describes the DNA loss due to Cas12a cleavage, while the second term is DNA increase from the polymerase reaction. The balance between these two terms decides the reaction fate – whether DNA will be amplified or entirely cleaved. This balance is particularly important at the initial phase of the CreDiT reaction when $C \approx C_0$ and $P \approx P_0$. Eq. (8) is approximated as

$$(9) \quad \frac{dD}{dt} \approx -\underbrace{\frac{k_c C_0}{k_{Mc}}}_{a_c} D + \underbrace{\frac{k_p P_0}{k_{Mp}}}_{a_p} D$$

where a_c and a_p are the reaction rates of *cis*-cleavage and polymerization, respectively.

For the Cas12a *cis*-cleavage, the reported kinetic values are $k_{1c} = 1.3 \times 10^7 \text{ M}^{-1} \text{ s}^{-1}$ (Ref. N1), $k_{2c} = 1.7 \times 10^{-3} \text{ s}^{-1}$ (Ref. N1), and $k_c = 5 \times 10^{-3} \text{ s}^{-1}$ (Refs. N2, N3), which gives $K_{Mc} = 5.2 \times 10^{-4} \mu\text{M}$. For the polymerase reaction, the RPA kit uses polymerase I, whose kinetic parameters are estimated to be $k_{1p} = 4 \times 10^7 \text{ M}^{-1} \text{ s}^{-1}$, $k_{2p} = 0.2 \text{ s}^{-1}$, and $k_p = 50 \text{ s}^{-1}$ (Ref. N4). These values give $K_{Mp} = 1.3 \mu\text{M}$. With $C_0 = 0.64 \mu\text{M}$ and $P_0 = 1.3 \mu\text{M}$ in our assay, the reaction rates are

	Enzyme concentration (μM)	Turnover rate (s^{-1})	MM constant (μM)	Reaction rate (s^{-1})
CAS12a cis	$C_0 = 0.64$	$k_c = 0.005$	$K_{Mc} = 5.2 \times 10^{-4}$	$a_c = 6.2$
Polymerase	$P_0 = 1.3$	$k_p = 50$	$K_{Mp} = 1.26$	$a_p = 52$

The DNA production is about 9-fold higher than the degradation by the Cas12a *cis*-cleavage activity. Based on the first-order kinetics (Eq. 9), the DNA amount will **exponentially increase over time**.

2. *Trans*-cleavage of primers

Activated Cas12a can interfere with DNA replication by indiscriminately cutting (*trans*-cleavage) RPA primers (single-stranded DNAs). However, kinetic considerations indicate that the polymerase reaction would dominate more than the Cas12a *trans*-cleavage reaction in CreDiT.

Following a similar procedure as in the *cis*-cleavage, the rate constant (a_t) of the *trans*-cleavage can be estimated as

$$(10) \quad a_t = \frac{k_t C_0}{K_{Mt}}$$

where k_t and K_{Mt} are the catalytic turnover rate and the MM constant for Cas12a *trans*-cleavage. We experimentally measured these values. Activated Cas12a (1 nM) was mixed with a substrate (fluorophore-quencher probe). The resulting fluorescent signals were monitored and converted into cleaved substrate concentrations (**Fig. N1a**). From these curves, the velocity (V) of the enzymatic reaction was obtained and fitted to an MM equation (**Fig. N1b**), which yielded $k_t = 1.2 \text{ s}^{-1}$ and $K_{Mt} = 3.4 \text{ }\mu\text{M}$.

Fig. N1. Kinetics of Cas12a *trans*-cleavage. (a) Temporal increase of cleaved substrate. Fluorophore-quencher probes (used in CreDiT) were used as a substrate. The concentration of the activated Cas12a was fixed (1 nM), while the substrate concentration was varied. (b) The velocity of the enzyme reaction was obtained from (a) and fitted to an MM equation. The fitting results yielded k_t and K_{Mt} . Data are from technical triplicate and displayed as mean \pm s.d.

The rate constants are now compared as

	Enzyme concentration (μM)	Turnover rate (s^{-1})	MM constant (μM)	Reaction rate (s^{-1})
CAS12a trans	$C_0 = 0.64$	$k_t = 1.2$	$K_{Mt} = 3.4$	$a_t = 0.2$
Polymerase	$P_0 = 1.3$	$k_p = 50$	$K_{Mp} = 1.3$	$a_p = 52$

Given this significant (>200-fold) difference in reaction rates, the polymerase is expected to outpace the Cas12a *trans*-cleavage activity, ensuring sufficient DNA amplification during the CreDiT assay.

References

- N1. Singh, D. et al. Real-time observation of DNA target interrogation and product release by the RNA-guided endonuclease CRISPR Cpf1 (Cas12a). *Proc. Natl. Acad. Sci. U S A* **115**, 5444-5449 (2018).
- N2. van Aelst, K., Martínez-Santiago, C. J., Cross, S. J. & Szczelkun, M. D. The Effect of DNA Topology on Observed Rates of R-Loop Formation and DNA Strand Cleavage by CRISPR Cas12a. *Genes (Basel)* **10**, 169 (2019).
- N3. Stella, S. et al. Conformational Activation Promotes CRISPR-Cas12a Catalysis and Resetting of the Endonuclease Activity. *Cell* **175**, 1856-1871.e21 (2018).
- N4. Kuznetsova, A. A., Fedorova, O. S. & Kuznetsov, N. A. Structural and Molecular Kinetic Features of Activities of DNA Polymerases. *Int. J. Mol. Sci.* **23**, 6373 (2022).

REVIEWERS' COMMENTS

Reviewer #1 (Remarks to the Author):

The authors have made a major corrections on the manuscript, and have addressed most of the concerns of the referees. But the problem for its publication in NC was that this topic about NA analysis with CRISPR has been wide reported, even with on-site analysis coupling with microfluidics. It seems that NC also published some papers on that topic. This work did not show any significant improvement, but a me-too, or me-better version.

Reviewer #4 (Remarks to the Author):

Thank you to the authors for their thorough responses. I am happy with the authors detailed responses and clarifying the points I raised. I would be happy with the paper getting published if these clarifications get added to the paper.

Responses to Reviewers' Comments

Reviewer 1

The authors have made a major corrections on the manuscript, and have addressed most of the concerns of the referees. But the problem for its publication in NC was that this topic about NA analysis with CRISPR has been wide reported, even with on-site analysis coupling with microfluidics. It seems that NC also published some papers on that topic. This work did not show any significant improvement, but a me-too, or me-better version.

We thank the reviewer for reevaluating our revised manuscript. We are glad to know that our responses addressed most of the concerns of the referees.

Reviewer 4

Thank you to the authors for their thorough responses. I am happy with the authors detailed responses and clarifying the points I raised. I would be happy with the paper getting published if these clarifications get added to the paper.1.

We convey our sincere gratitude to the reviewer for this encouraging comment. We have incorporated all our responses, including those to Reviewer 4's comments, in the revised manuscript.